# HEAD2TOE: UTILIZING INTERMEDIATE REPRESENTATIONS FOR BETTER OOD GENERALIZATION

## ABSTRACT

Transfer-learning methods aim to improve performance in a data-scarce target domain using a model pretrained on a data-rich source domain. A cost-efficient strategy, *linear probing*, involves freezing the source model and training a new classification head for the target domain. This strategy is outperformed by a more costly but state-of-the-art method—*fine-tuning* all parameters of the source model to the target domain—possibly because fine-tuning allows the model to leverage useful information from intermediate layers which is otherwise discarded by the later pretrained layers. We explore the hypothesis that these intermediate layers might be directly exploited by linear probing. We propose a method, *Head-to-Toe probing* (HEAD2TOE), that selects features from all layers of the source model to train a classification head for the target-domain. In evaluations on the Visual Task Adaptation Benchmark (VTAB), Head2Toe matches performance obtained with fine-tuning on average, but critically, for out-of-distribution transfer, Head2Toe outperforms fine-tuning.

## 1 INTRODUCTION

Tranfer learning is a widely used method for obtaining strong performance in a variety of tasks where training data is scarce—see Zhu et al. (2020); Alyafeai et al. (2020); Zhuang et al. (2020) for recent application-specific surveys. A well-known recipe for transfer learning involves the supervised or unsupervised pretraining of a model on a *source* task with a large training dataset (also referred to as *upstream training*). After pretraining, the model's output head is discarded, and the rest of the network is used to obtain a feature embedding, i.e., the output of what was formerly the penultimate layer of the network. When transferring to a *target* task, a new output head is trained on top of the feature extractor (*downstream training*). This approach makes intuitive sense: if a linear combination of embedding features performs well on the source task, we expect a different linear combination of features to generalize to the target domain, provided the source and target tasks are similar.

This approach of training a new output head, referred to as *linear probing (*LINEAR*)*, often yields significant improvements in performance on the target task over training the network from scratch (Kornblith et al., 2019). An alternative to LINEAR is *fine-tuning (*FINETUNING*)*, which uses target-domain data to adapt all weights in the feature extractor together with the new output head. This procedure requires doing forward and backward passes through the entire network at each training step and therefore its per-step cost is significantly higher than LINEAR. Furthermore, since the entire network is fine-tuned, the entire set of new weights needs to be stored for every target task, making FINETUNING impractical when working on edge devices or with a large number of target tasks. However, FINETUNING is often preferred over LINEAR since it consistently leads to better performance on a variety of target tasks even when data is scarce (Zhai et al., 2019).

FINETUNING's superior generalization in the low-data regime is counterintuitive given that the number of model parameters to be adapted is often large relative to the amount of available training data. How does FINETUNING learn from few examples successfully? We conjecture that FINETUNING better leverages existing internal representations rather than discovering entirely new representations; FINETUNING exposes existing features buried deep in the net for use by the classifier. Under this hypothesis, *features needed for transfer are already present in the pretrained network and might be identified directly without fine-tuning the backbone itself.*

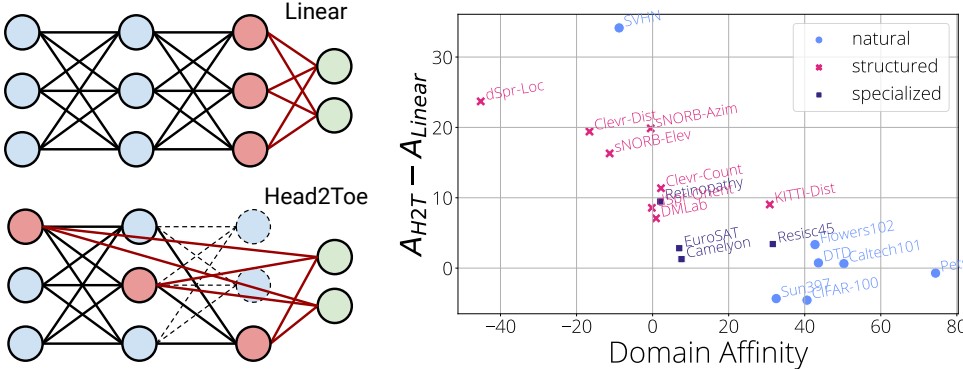

Figure 1: **(left)** HEAD2TOE selects the most useful features from the network and trains a linear head on top. **(right)** HEAD2TOE improves OOD generalization. Domain affinity refers to the difference between linear probe and scratch accuracies (defined in Section 2) and the y-axis represents the improvements (delta accuracy) made by HEAD2TOE over linear probe.

In this work, we propose and explore methods for selecting useful features embedded in internal layers of a pretrained net, concatenating them to the embedding produced by the pretrained net, and then applying the LINEAR transfer approach to the augmented representation (Fig. 1-left). Our approach leads to significant improvements over LINEAR as shown in Fig. 1-right. Our key contributions are as follows:

1. We perform analyses to better understand two factors that affect the benefit of incorporating intermediate representations: feature selection and the degree to which a target domain is out-of-distribution (OOD).

2. We introduce head-to-toe probing (HEAD2TOE), a simple recipe for selecting relevant features from intermediate representations for few-shot generalization. On the VTAB collection of data sets, we show that HEAD2TOE outperforms LINEAR and matches the performance of more computationally costly FINETUNING.

3. Critically, HEAD2TOE outperforms FINETUNING on OOD target domains. If a practitioner can make an educated guess about whether a target domain is OOD with respect to a source, using HEAD2TOE improves on the state of the art for transfer learning

## 2 PRELIMINARIES

**Source domain and backbone models.** In our experiments, we use source models pretrained on ImageNet-2012 (Russakovsky et al., 2015), a large scale image classification benchmark with 1000 classes and over million natural images. We benchmark HEAD2TOE using convolutional (ResNet-50, Wu et al., 2018) and attention-based (ViT-B/16, Dosovitskiy et al., 2021) architectures pretrained on ImageNet-2012.

**Target domains.** In this work, we focus on target tasks with few examples (i.e., *few-shot*) and use Visual Task Adaptation Benchmark-1k (Zhai et al., 2019) to evaluate different methods. Visual Task Adaptation Benchmark-1k consists of 19 different classification tasks, each having between 2 to 397 classes and a total of 1000 training instances. The domains are grouped into three primary categories: (1) natural images (*natural*), (2) specialized images using non-standard cameras (*specialized*), and (3) rendered artificial images (*structured*).

**Characterizing out-of-distribution domains.** Consider the relationship between source and target domains. If the domain overlap is high, then features extracted for linear classification in the source domain should also be relevant for linear classification in the target domain, and LINEAR should yield performance benefits. If the domain overlap is low, then constraining the target domain to use the source domain embedding may be harmful relative to training a model from scratch on

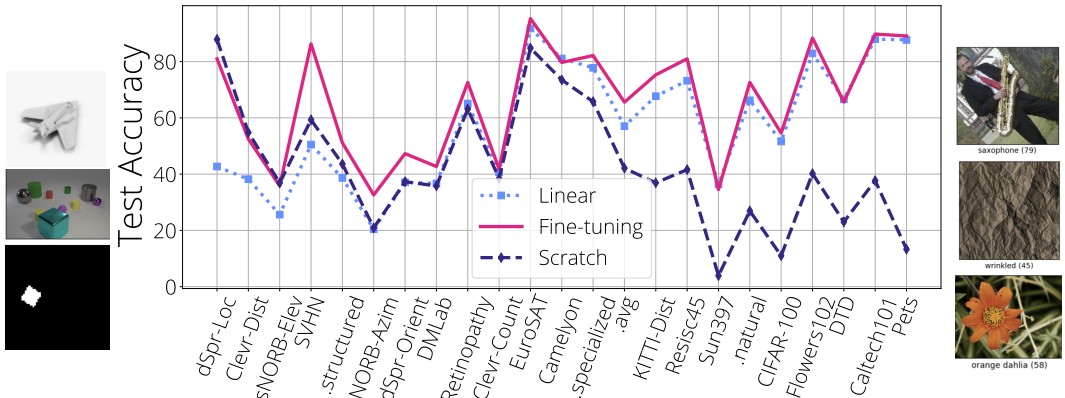

Figure 2: **Characterizing out-of-distribution domains** Generalization performance of various baselines on the VTAB-1k benchmark using ResNet-50 architecture and 224 image size. The architecture is pretrained on ImageNet-2012 for the transfer learning (TL) baselines. Datasets (and the three groups of the benchmark) are ordered according to their Domain Affinity scores in ascending order from left to right. Examples from the left- and right-most datasets are also shown on corresponding sides.

the target domain. Therefore, we might quantify the source-target distribution shift in terms of how beneficial LINEAR is relative to training a model from scratch (denoted SCRATCH):

$$DomainAffinity = \text{Acc}_{\text{LINEAR}} - \text{Acc}_{\text{SCRATCH}}.$$

In Fig. 2, we sort the 19 VTAB-1k target tasks by their domain affinity to ImageNet-2012—from low to high—for a pretrained ResNet-50 backbone. The Figure shows examples of the three target domains with the most and least distribution shift (left and right sides, respectively). These examples seem consistent with intuitive notions of distribution shift.

**Baselines.** Fig. 2 also presents transfer test accuracy of LINEAR, FINETUNING, and SCRATCH baselines. Consistent with the literature, FINETUNING performs as well or better than the other two baselines. For in-distribution targets (right side of graph), LINEAR matches FINETUNING; for OOD targets (left side of graph), LINEAR is worse than FINETUNING. With distribution mismatch, the source network may filter information available in lower layers because it is not needed for the source task but is essential for the target task. Observing FINETUNING performs better than SCRATCH even in OOD tasks, we hypothesize that intermediate features are key for the FINETUNING, since if learning novel features was possible using limited data, training the network from scratch would work on-par with FINETUNING. Motivated by this observation, HEAD2TOE probes the intermediate features of a network directly and aims to eliminate the need for fine-tuning.

## 3 HEAD2TOE UTILIZATION OF PRETRAINED BACKBONES

### 3.1 YOUR REPRESENTATIONS ARE RICHER THAN YOU THINK

In this section, we conduct a simple experiment to demonstrate the potential of using representations from intermediate layers. We concatenate the feature embedding of a pretrained ResNet-50 backbone (features from the penultimate layer) with features from *one* additional layer and train a linear head on top of the concatenated features. When extracting features from convolutional layers, we reduce the dimensionality of the convolutional stack of feature maps using strided average pooling, with a window size chosen so that the resulting number of features is similar to the number of embedding features (2048 for ResNet-50).

In order to estimate an upper bound on how much the performance can be improved over LINEAR by including a single intermediate layer, we use an oracle to select the layer which yields the largest boost in test performance (for each target task separately). Percentage improvement over LINEAR using this ORACLE is shown in Fig. 3-left. We observe a negative correlation (-0.745, Spearman)

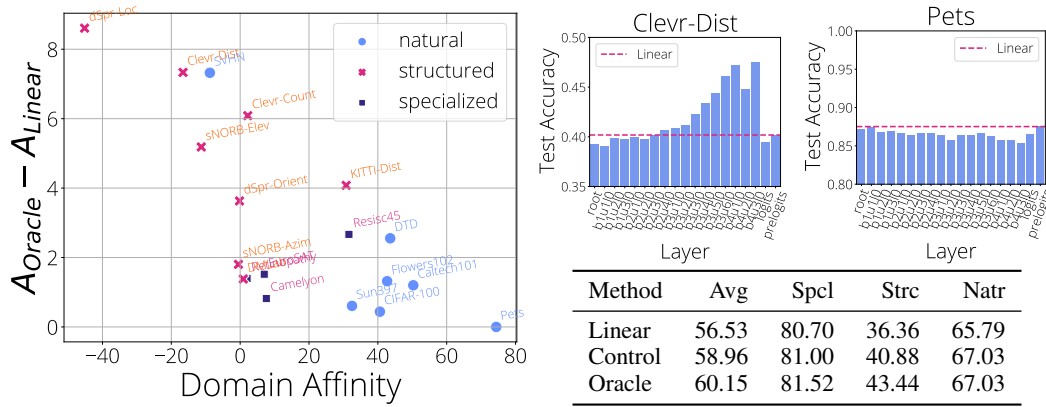

Figure 3: **(left)** Accuracy gains when prelogits are augmented with an additional layer correlates negatively (-0.745, Spearman) with domain affinity. **(right-top)** effect of using features from intermediate layers for Clevr-Dist (low domain affinity) and Pets (high domain affinity) tasks **(right-bottom)** Test accuracies of various baselines on VTAB-1k. *Linear* uses only prelogits, *Oracle* averages are obtained by using the layer that gives best generalization (test) for each task. *Control* experiment uses a second feature embedding from a second pretrained network trained using a different seed. We use ResNet-50 models pretrained on ImageNet-2012.

between the domain affinity of a target task and the accuracy gain. As predicted earlier, adding intermediate representations does not improve in-domain generalization because feature embedding already contains the most useful features. In contrast, generalization on out-of-domain tasks are improved significantly when intermediate features are used.

In Fig. 3-top-right, we show test accuracy as a function of the ResNet-50 layer whose internal representation is used to augment the feature embeddings to boost LINEAR transfer. Figures for remaining tasks can be found in Appendix D. Different tasks on VTAB benchmark prefer inclusion of different layers to obtain the best generalization accuracy, emphasizing the importance of selecting the right set of features for optimal generalization. Overall, the ORACLE that selects the layer with best test performance for each task yields an average of 3.5% improvement on VTAB-1k benchmark. One possible explanation for the improvement in performance with the augmented representation is simply that it has more degrees of freedom (4096 features instead of 2048). To demonstrate that the improvement is due to inclusion of intermediate layers and not simply due to increased dimensionality, we formed a 4096-element feature representation by using a second feature embedding obtained from a second ResNet-50 backbone pretrained on ImageNet-2012 (CONTROL). Note that this experiment bears similarity to *ensembling* (Zhou et al., 2002), which is known to bring significant accuracy gains on its own (Mustafa et al., 2020). Utilizing a second backbone doubles the amount of resources required, yet it falls 1% shy of the Oracle performance demonstrating the extent to which using intermediate representations can be helpful for generalization.

## 3.2 HEAD2TOE

Motivated by our observations in the previous section, we hypothesize that we can attain—or possibly surpass—the performance of FINETUNING without changing the backbone itself by using LINEAR augmented with well-chosen intermediate activations. However, this approach introduces the problem of selecting features. The number of possible features to include is large relative to the amount of training data available, leading to the potential of overfitting. In this section, we explore various regularization and feature selection methods, which we then incorporate into our method, HEAD2TOE.

**Notation.** Our method applies to any network with any type of layers, but here, for simplicity and without loss of generality, we consider a network with $L$ fully connected layers, each layer receiving input from the layer below:

$$\boldsymbol{z}_\ell = \boldsymbol{h}_{\ell-1}\boldsymbol{W}_\ell \quad ; \quad \boldsymbol{h}_\ell = f(\boldsymbol{z}_\ell) \tag{1}$$

where the subscript denotes a layer index, $h_0 = x$ is the input, $f$ is the activation function, $W_\ell$ is the weight matrix of layer $\ell$, and $z_L$ is the logit vector used for classification. When transferring a pretrained network to a target task using LINEAR, we discard the last layer of the pretrained network and train a new set of linear weights, $W_L'$, such that predictions (logits) for the new task are obtained by $z_L' = h_{L-1}W_L'$.

**Head2Toe.** Consider a simple scheme that augments the backbone embedding with activations from all layers of the network, such that:

$$z_L' = h_{all}W_{all} \quad ; \quad h_{all} = \text{concat}(a_1(h_1), a_2(h_2), ..., a_L(h_L)) \tag{2}$$

where $a_\ell(.)$ denotes a fixed function to reduce the dimensionality of the activation vector at a given layer $\ell$. Such functions become valuable when considering network architectures that generate large number of intermediate features. For example, for convolutional networks, where the representations at each layer are structured as multi-channel spatial arrays of detectors, we perform two-dimensional average pooling as explained in Section 3.1 and Section 5. Even with dimensionality reduction, $h_{all}$ can exceed a million elements, and $W_{all}$ is underconstrained by the training data, leading to overfitting. Further, without care $W_{all}$ may become so large as to be impractical for deploying this model.[1]. We can address these issues by selecting a subset of features before training the target-domain classifier. Equivalently, we can zero out the non-selected rows of $W_{all}$. We will show that appropriate selection of a subset of features leads to better generalization than using all features.

**Feature selection based on group lasso.** Group lasso (Yuan & Lin, 2006) is a popular method for selecting relevant features in multi-task adaptation settings (Argyriou et al., 2007; Nie et al., 2010). When used as a regularizer on a weight matrix $W$, the group-lasso regularizer encourages the $\ell_2$ norm of the rows of the matrix to be sparse, and is defined as:

$$|W|_{2,1} = |s|_1 = \sum_i |s_i| \quad ; \quad s_i = \sqrt{\sum_j w_{ij}^2} \tag{3}$$

To determine which features are most useful for the task, the linear head is trained with group-lasso regularization on $W_{all}$. In contrast to the approach taken by Argyriou et al. (2007) and Nie et al. (2010), which use costly matrix inversions, we incorporate the regularization as a secondary loss and train with stochastic gradient descent. Following training, the $\ell_2$ norms of the rows of $W_{all}$, $s$, are used to decide which features are most relevant. We refer to $s_i$ as the *relevance score* for feature $i$. We pick a fraction $F$ of all features with the greatest relevance and train a new linear head to obtain the final logit mapping. Feature selection alone provides strong regularization, therefore during the final training we don't use any additional regularization.

We make two comments on this. First, because the initial round of training $W_{all}$ with the group-lasso regularizer is used only to rank features by importance, the method is quite robust to the regularization coefficient; it simply needs to be large enough to distinguish the contribution of the individual features. Second, interpreting $s_i$ as the importance of feature $i$ depends on all features having the same dynamic range. This constraint will often be satisfied naturally due to the normalization layers found in the architecture itself (we read out representations after normalization). To cover the remaining cases (where no normalization exists between two layers) we normalize the representations of each layer to a unit norm.

**Selection of $F$ and the cost of HEAD2TOE.** The fraction $F$ determines the total number of features retained. One would expect the optimal value to depend on the target task. We select $F$ based on validation performance. This validation procedure is inexpensive compared to the cost of the initial phase of the algorithm (i.e., training of $W_{all}$ to obtain $s$) due to the reduced number of features in the second step. Overall, HEAD2TOE together with its validation procedure requires 18% more operations compared to training $W_{all}$ alone (details shared in Appendix B).

**Verifying HEAD2TOE.** In Fig. 4-left, we demonstrate the effectiveness of group lasso on identifying relevant intermediate features of a ResNet-50 for transfer from ImageNet. We rank all features

---

[1]For example, using a pooling size of 2, ResNet-50 generates 1.7 million features and storing $W_{all}$ requires 6.6e8 parameters (2.6GB for float32) for SUN-397.

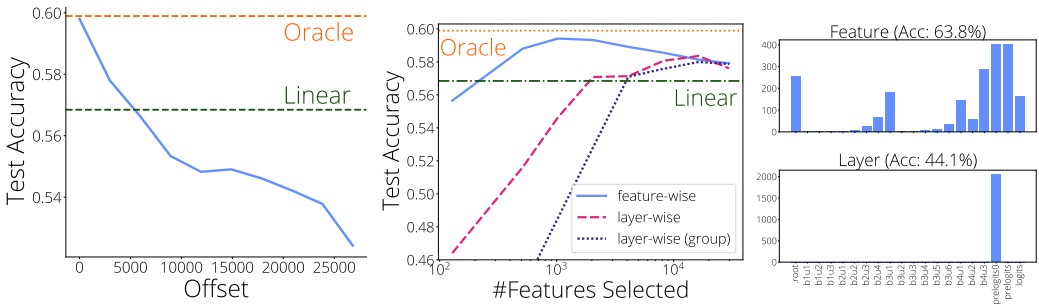

Figure 4: **(left)** Over all VTAB tasks, average accuracy of HEAD2TOE when selecting 2048 consecutive features sorted by their relevance score, starting with an index specified by *offset*. **(center)** Average accuracy over all VTAB tasks as a function of the number of features included. In this experiment, we only use features from the first layer of each of the 18 ResNet blocks and adjust pooling to have around 2048 features for each layer, totalling 29800 features. Results show that selecting layers performs worse than selecting features when adapting to a target domain. **(right)** Distribution of 2048 intermediate features retained from a ResNet-50 when using feature-wise and layer-wise scores on the SVHN transfer task.

by their relevance score, $s_i$, and we select groups of 2048 consecutive features beginning at a particular offset in this ranking. Offset 0 corresponds to selecting the features with largest relevance. We calculate average test accuracies across all VTAB tasks. As the figure shows, test accuracy decreases monotonically with the offset, indicating that the relevance score predicts the importance of including a feature in the linear classifier.

HEAD2TOE selects individual features independent of the layer in which they are embedded. We compare this *feature-wise* strategy to selecting layers as whole (i.e., selecting all features in a layer or none). One might expect *layer-wise* selection to be superior because it involves fewer decisions and therefore less opportunity to overfit to the validation set that is being used for selection. Further, layer-wise selection may be superior because the relevance of one feature in a layer may indicate the relevance of others. To obtain a layer-wise relevance score, we compute the mean relevance score of all features in a layer and then rank layers by relevance. We also run an alternative layer selection algorithm, in which we group weights originating from the same layer together and select layers using the $\ell_2$ norm of the groups (*layer-wise (group)*). Fig. 4-center compares feature-wise and layer-wise selection methods, matched for number of features retained. Feature-wise selection performs better than layer-wise selection on average and the best performance is obtained when around 1000 features are kept. We share figures for all 19 datasets in Appendix E. Fig. 4-right shows the distribution of features selected over different layers for both strategies.

Note that HEAD2TOE's use of a fixed backbone means that as we search for features to include, the extracted features themselves are not changing. Consequently, we can calculate them once and reuse as necessary, instead of recalculating at every step, as required for FINETUNING. Furthermore, since the backbone is frozen, the only additional weight storage required for each target task is the output head. Such properties are key for transfer-learning methods to be practical when using models with billions of parameters.

## 4 RELATED WORK

**Transfer Learning** is studied extensively in the literature and used widely in practice. Most similar to our work is ELMo (Peters et al., 2018), which averages intermediate representations of a language model (2 LSTM embeddings) using a learned linear combination (a softmax). ELMo requires embeddings to be same size and is most similar to the suboptimal layer-selection baseline shown in Fig. 4-center. Peters et al. (2019) compared FINETUNING with LINEAR and reported a small gap in performance for ELMo and BERT models. Kornblith et al. (2019) showed that ImageNet-2012 performance correlates highly with LINEAR and FINETUNING. Using a large pool of pretrained models, Renggli et al. (2020) examined various metrics to decide which backbone to transfer. Given the ever increasing size and never saturating performance of pretrained models, the importance of reducing the cost of FINETUNING models is stated in the literature regularly. Methods like feature-wise transformations (Bilen & Vedaldi, 2017), residual adapters (Houlsby et al., 2019; Rebuffi et al.,

2017), Diff-pruning (Guo et al., 2021) and selective fine-tuning (Guo et al., 2019; Fu et al., 2021) are proposed in order to reduce the cost of storing fine-tuned models. Zhang et al. (2021) showed that selecting a sub-network of a pretrained model can increase OOD generalization. However, none of these methods match the simplicity of training (and storing) a linear classifier. Teerapittayanon et al. (2016), Kaya et al. (2019), and Zhou et al. (2020) studied intermediate representations to reduce "overthinking" and thus provide better early-exit criteria. Similarly Baldock et al. (2021) showed a correlation between early classification of a sample and how easy its classification is.

**Alternatives** to transfer learning exist to learn from few labeled examples. Meta-learning (Thrun & Pratt, 1998; Schmidhuber, 1987; Schmidhuber et al., 1997) is used (among other things) to tackle the few-shot learning problem (i.e learning from limited data) from a learning-to-learn perspective; see Hospedales et al. (2020) for a comprehensive survey. In-between the representation learning and learning-to-learn paradigms lie approaches that use multi-domain training as an inductive bias, either without (Dvornik et al., 2020; Triantafillou et al., 2021; Li et al., 2021b;a) or with (Liu et al., 2021) meta-learning. However, in the large-scale setting FINETUNING remains a top-performing approach to few-shot classification (Dumoulin et al., 2021).

**Feature selection** aims to find the most relevant features from the input and is studied extensively in the machine-learning literature. Approaches can be grouped according to whether labeled data is used—supervised (Nie et al., 2010) or unsupervised (He et al., 2005; Balın et al., 2019; Atashgahi et al., 2020)—or what high-level approach is taken—filter methods (Blum & Langley, 1997), wrapper methods (Kohavi & John, 1997), or embedded methods (Yuan & Lin, 2006). Most relevant to our work are embedded supervised methods as they have good scaling properties while achieving the best performance which is vital in our setting with over a million features. Embedded supervised feature selection methods use a cost function to iteratively refine the subset of features selected and popular approaches include forward selection (Viola & Jones, 2001; Borboudakis & Tsamardinos, 2019), backward selection (pruning) (Mozer & Smolensky, 1989; Guyon et al., 2004) and regularization/feature-ranking based methods (Yuan & Lin, 2006; Blum & Langley, 1997; Zhao et al., 2010). Most relevant to our work is Argyriou et al. (2007); Nie et al. (2010), both of which uses $\ell_{2,1}$ regularization to select features, however their approach requires matrix inversions which is not practical in our setting. We point interested readers to the survey of Gui et al. (2017) and book of Boln-Canedo et al. (2015) for a detailed discussion on the topic.

## 5 EVALUATING HEAD2TOE

We evaluate HEAD2TOE on the VTAB-1k benchmark using two popular vision architectures, ResNet-50 (Wu et al., 2018) and ViT-B/16 (Dosovitskiy et al., 2021), both pretrained on ImageNet-2012. ResNet-50 consists of 50 convolutional layers. To utilize the intermediate convolutional features, we aggregate spatial information using average pooling on non-overlapping patches, as explained in Section 3. We adjust the pooling size to target a fixed dimensionality of the representation. For example, for a feature map of shape $20 \times 20 \times 128$ and a *target size* of 512, we average-pool disjoint patches of size $5 \times 5$, resulting in 4 features per channel and 1024 features in total. This helps us to balance different layers in terms of number features they contribute to the concatenated embedding. We normalize features coming from each layer to a unit norm. This scaling preserves the relative magnitude of features within a layer while accounting for inter-layer differences and works better than normalizing each feature separately or not normalizing at all. ViT-B/16 consists of 12 multi-headed self-attention layers. When we aggregate the output of the self-attention layers, we perform 1-D average pooling over the patch/token dimension choosing the pooling size to match a target number of features as before. Given that token dimension is permutation invariant, 1-D average pooling might not be the best choice here and better aggregation function can provide further gains. We share a detailed list of intermediate representations utilized for each architecture in Appendix H.

HEAD2TOE selects a subset of features and trains a linear classifier without regularization on top of the selected features. We compare HEAD2TOE with regularization baselines that utilize all features without doing feature selection. These baselines are denoted as +All-$\ell_1$, +All-$\ell_2$ and +All-$\ell_{2,1}$ according to the norm they use.

We perform five-fold cross validation for each task and method in order to pick the best hyper-parameters. All methods search over the same learning rates and training steps (two values of

| | Natural | | | | | | | Specialized | | | | Structured | | | | | | | | |
|---|---|---|---|---|---|---|---|---|---|---|---|---|---|---|---|---|---|---|---|---|
| | CIFAR-100 | Caltech101 | DTD | Flowers102 | Pets | SVHN | Sun397 | Camelyon | EuroSAT | Resisc45 | Retinopathy | Clevr-Count | Clevr-Dist | DMLab | KITTI-Dist | dSpr-Loc | dSpr-Ori | sNORB-Azim | sNORB-Elev | Mean |
| Linear | 48.5 | 86.0 | 67.8 | 84.8 | 87.4 | 47.5 | 34.4 | **83.2** | 92.4 | 73.3 | 73.6 | 39.7 | 39.9 | 36.0 | 66.4 | 40.4 | 37.0 | 19.6 | 25.5 | 57.0 |
| +All-$\ell_2$ | 44.7 | 87.0 | 67.8 | 84.2 | 86.1 | 81.1 | 31.9 | 82.6 | 95.0 | 76.5 | 74.5 | 50.0 | 56.3 | 38.3 | 65.5 | 59.7 | 44.5 | 37.5 | 40.0 | 63.3 |
| +All-$\ell_1$ | 50.8 | 88.6 | 67.4 | 84.2 | 87.7 | 84.2 | 34.6 | 80.9 | 94.9 | 75.6 | **74.7** | 49.9 | 57.0 | 41.8 | 72.9 | 59.0 | 44.8 | 37.5 | 40.8 | 64.6 |
| +All-$\ell_{2,1}$ | 49.1 | 86.7 | **68.5** | 84.2 | 88.0 | 84.4 | **34.8** | 81.5 | 94.9 | 75.7 | 74.3 | 48.3 | 58.4 | 42.0 | 74.4 | 58.8 | 45.2 | 37.8 | 34.4 | 64.3 |
| Head2Toe | 47.1 | 88.8 | 67.6 | 85.6 | 87.6 | 84.1 | 32.9 | 82.1 | 94.3 | 76.0 | 74.1 | **55.3** | **59.5** | **43.9** | 72.3 | 64.9 | **51.1** | **39.6** | **43.1** | **65.8** |
| Fine-tuning* | **54.6** | **89.8** | 65.6 | **88.4** | **89.1** | **86.3** | 34.5 | 79.7 | **95.3** | **81.0** | 72.6 | 41.8 | 52.5 | 42.7 | **75.3** | 81.0 | 47.3 | 32.6 | 35.8 | 65.6 |
| Scratch* | 11.0 | 37.7 | 23.0 | 40.2 | 13.3 | 59.3 | 3.9 | 73.5 | 84.8 | 41.6 | 63.1 | 38.5 | 54.8 | 35.8 | 36.9 | **87.9** | 37.3 | 20.9 | 36.9 | 42.1 |

| | Natural | | | | | | | Specialized | | | | Structured | | | | | | | | |
|---|---|---|---|---|---|---|---|---|---|---|---|---|---|---|---|---|---|---|---|---|
| | CIFAR-100 | Caltech101 | DTD | Flowers102 | Pets | SVHN | Sun397 | Camelyon | EuroSAT | Resisc45 | Retinopathy | Clevr-Count | Clevr-Dist | DMLab | KITTI-Dist | dSpr-Loc | dSpr-Ori | sNORB-Azim | sNORB-Elev | Mean |
| Linear | 55.0 | 81.0 | 53.6 | 72.1 | 85.3 | 38.7 | 32.3 | 80.1 | 90.8 | 67.2 | 74.0 | 38.5 | 36.2 | 33.5 | 55.7 | 34.0 | 31.3 | 18.2 | 26.3 | 52.8 |
| +All-$\ell_2$ | 57.3 | 87.0 | 64.3 | 82.8 | 84.0 | 75.7 | 32.4 | 82.0 | 94.7 | 79.7 | **74.8** | 47.4 | 57.8 | 41.4 | 62.8 | 46.6 | 33.3 | 31.0 | 38.8 | 61.8 |
| +All-$\ell_1$ | 58.4 | **87.3** | **64.9** | 83.3 | 84.6 | 80.0 | 34.4 | 82.3 | **95.6** | 79.6 | 73.6 | 47.9 | 57.7 | 42.2 | 65.1 | 44.5 | 33.4 | 32.4 | 38.4 | 62.4 |
| +All-$\ell_{2,1}$ | 59.6 | 87.1 | 64.9 | 85.2 | 85.4 | 79.5 | 35.3 | 82.0 | 95.3 | 80.6 | 74.2 | 47.9 | 57.8 | 40.7 | 64.9 | 46.7 | 33.6 | 31.9 | 39.0 | 62.7 |
| Head2Toe | 58.2 | 87.3 | 64.5 | **85.9** | 85.4 | 82.9 | 35.1 | 81.2 | 95.0 | 79.9 | 74.1 | 49.3 | 58.4 | 41.6 | 64.4 | 53.3 | 32.9 | **33.5** | **39.4** | 63.3 |
| Fine-tuning | 62.6 | 83.1 | 61.5 | 80.4 | 86.8 | 83.0 | 33.7 | **83.2** | 94.9 | 78.0 | 73.6 | 58.3 | 59.2 | **43.6** | 39.8 | 61.4 | **44.8** | 27.1 | 26.2 | 62.2 |
| Fine-tuning+ | **65.2** | 82.1 | 60.5 | 85.2 | **86.9** | **86.7** | **36.9** | 82.4 | 95.2 | **81.3** | 73.0 | **68.2** | **60.6** | 41.7 | **73.9** | **69.3** | 28.0 | 25.7 | 36.1 | **65.2** |

Table 1: Median test accuracy over 3 seeds on the VTAB-1k benchmark using pretrained (**top**) ResNet-50 and (**bottom**) ViT-B/16 backbones. Regularization baselines that use all layers are indicated with the +*All* prefix. "*" indicates results obtained from Zhai et al. (2019). Fine-tuning results for ViT-B/16 are obtained using code and checkpoints provided by Dosovitskiy et al. (2021).

each). Methods that leverage intermediate features (i.e., regularization baselines and HEAD2TOE) additionally search over regularization coefficients and the size of the aggregated representation at each layer. The FINETUNING baseline searches over 4 hyper-parameters; thus the comparison of HEAD2TOE, which searches over 24 values, to fine-tuning might seem unfair. However, this was necessary due to fine-tuning being significantly more costly than training a linear classifier, even with intermediate features. We repeat each evaluation using 3 different seeds and report median values and share standard deviations in Appendix C. More details on hyper-parameter selection and best values can be found in Appendix A.

## 5.1 RESNET-50

The top half of Table 1 presents results on the 19 VTAB-1k target domains when transferring from a pretrained ResNet-50 architecture. On average, HEAD2TOE slightly outperforms all other methods, including FINETUNING (see rightmost column of Table). HEAD2TOE, along with the regularization baselines that use intermediate layers, is far superior to LINEAR, indicating the value of the intermediate layers. And HEAD2TOE is superior to the regularization baselines, indicating the value of explicit feature selection. Among the three categories of tasks, HEAD2TOE excels relative to the other methods for the *specialized* category, but does not outperform FINETUNING for the *natural* and *structured* categories.

Does HEAD2TOE select different features for each task? Which layers are used more frequently? In Appendix F, we show the distribution of features selected across different layers and the amount of intersection among features selected for different tasks and seeds. We observe a high variety among layers and features motivating the importance of performing feature selection for each task separately. HEAD2TOE exceeds FINETUNING performance, but requires only 0.5% of FLOPs during training on average. Similarly, the cost of storing the adapted model is reduced to 1% on average. We discuss HEAD2TOE's computational and storage costs in detail in Appendix B.

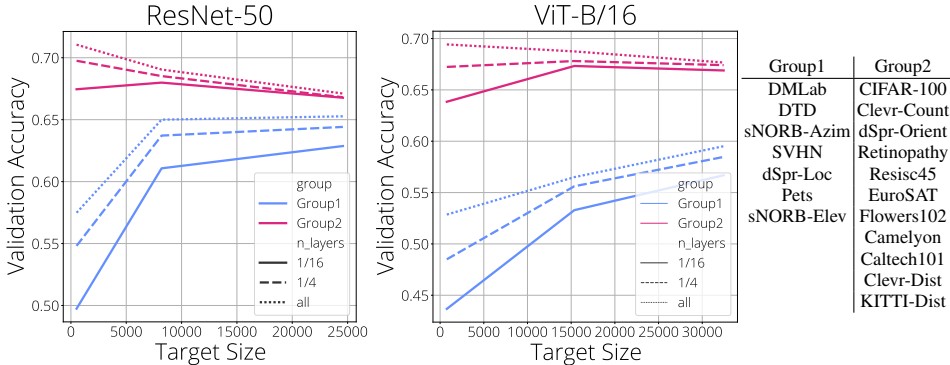

Figure 5: Effect of increasing the number of intermediate features that HEAD2TOE uses from the **(left)** ResNet-50 backbone and **(right)** ViT-B/16 backbone. The abscissa of the graph indicates the dimensionality of the representation extracted from each layer of the backbone (*target size*). The tasks are split into two groups (see right side of Figure), which show different behavior. The solid, dashed, and dotted lines indicate the fraction of layers selected for forming the representation used by HEAD2TOE: 1/16, 1/4, and 1, respectively. Scaling curves for individual tasks can be found in Appendix G.

## 5.2 VIT-B/16

Results for ViT-B/16 are shared in the bottom half of Table 1. As with the ResNet-50 architecture, HEAD2TOE achieves the best accuracy among methods that keep the backbone fixed: HEAD2TOE improves accuracy over LINEAR by about 10% on average (in absolute performance), and HEAD2TOE outperforms the regularization baselines that include intermediate features but that do not explicitly select features. The story for HEAD2TOE versus FINETUNING is a bit more complicated. The FINETUNING recipe used by Dosovitskiy et al. (2021) includes many tweaks such as selective data-augmentation, image upsampling, learning rate warm-up, and gradient clipping. In order to make a direct comparison with LINEAR, in the next-to-last row of the Table we report the performance of FINETUNING without all of these tweaks; HEAD2TOE obtains about a 1% absolute accuracy improvement over FINETUNING in this setting. However, when some of the tweaks are incorporated into FINETUNING—in particular gradient clipping—FINETUNING achieves a 3% improvement over HEAD2TOE. Since it is a linear classifier, adding gradient clipping to HEAD2TOE doesn't improve the results. That being said, HEAD2TOE retains its competitive advantage in terms of resource costs and its accuracy would likely similarly benefit from the development of more sophisticated tricks (notably moving beyond the naive 1-D pooling of ViT features), much like FINETUNING has gained over time.

## 5.3 SCALING HEAD2TOE

In Fig. 5 we vary the number of intermediate features used for each of our pretrained backbones. We observe that including all layers always performs better on average. However when varying the number of target features for each layer, we observed 2 distinct sub-group of target tasks that behave differently as the number of features increases. This observation informed our discussion to include both small and large target sizes in our validation hyper parameter search.

## 6 CONCLUSION

In this work, we introduced HEAD2TOE, an approach that extends linear probing (LINEAR) by selecting the most relevant features among a pretrained network's intermediate representations. We show that doing so greatly improves performance over LINEAR and allows the approach to reach a performance competitive with—and in some cases superior to—FINETUNING. Our findings challenge the conventional belief that FINETUNING is required to achieve good performance on OOD tasks. While more work is needed before HEAD2TOE can realize the full computational benefits of linear probing, our work paves the way for applying new and more efficient feature selection approaches and for experimenting with HEAD2TOE probing in other domains such as regression, video classification, object detection, reinforcement learning, and language modelling domains.

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

## A    VALIDATION PROCEDURE FOR DIFFERENT ALGORITHMS

We perform hyper-parameter validation for each VTAB task separately. For all methods, we chose the learning rate and the total number of training steps using the grid $lr = 0.1, 0.01$ and $steps = 500, 5000$, following the lightweight hyper-parameter sweep recommended by the VTAB benchmark (Zhai et al., 2019).

For regularization baselines $\ell_1$, $\ell_2$ and $\ell_{2,1}$ we search for regularization coefficients using $(0.00001, 0.0001, 0.001)$. We include an extra value in this setting in order to account for the overhead introduced by HEAD2TOE.

For HEAD2TOE we choose $\ell_{2,1}$ regularization coefficients from $(0.001, 0.00001)$ and target feature sizes from $(1024, 16384, 40000)$ for ResNet-50 and $(768, 15360, 32448)$ for ViT-B/16. After calculating feature scores HEAD2TOE validates the following fractions: $(0.0005, 0.001, 0.002, 0.005, 0.01, 0.02, 0.05, 0.1)$ and thus requires 18% more operations compared to other regularization baselines. Note that this is because initial training to obtain feature scores $s_i$ is performed once and therefore searching for optimal number of features has a small overhead. Hyper parameters selected by HEAD2TOE for each VTAB task are shared in Table 2 and Table 3. Next we explain how this overhead is estimated.

| Dataset | Target Size | F | LR | Steps | $\ell_{2,1}$ Coef. |
|---|---|---|---|---|---|
| Caltech101 | 8192 | 0.010 | 0.01 | 5000 | 0.00001 |
| CIFAR-100 | 512 | 0.200 | 0.01 | 500 | 0.00001 |
| Clevr-Dist | 8192 | 0.001 | 0.01 | 500 | 0.00100 |
| Clevr-Count | 512 | 0.005 | 0.10 | 5000 | 0.00100 |
| Retinopathy | 8192 | 0.200 | 0.01 | 500 | 0.00001 |
| DMLab | 8192 | 0.020 | 0.01 | 500 | 0.00001 |
| dSpr-Orient | 512 | 0.200 | 0.01 | 500 | 0.00001 |
| dSpr-Loc | 8192 | 0.005 | 0.10 | 500 | 0.00100 |
| DTD | 24576 | 0.005 | 0.01 | 5000 | 0.00001 |
| EuroSAT | 512 | 0.100 | 0.01 | 500 | 0.00001 |
| KITTI-Dist | 8192 | 0.020 | 0.01 | 500 | 0.00001 |
| Flowers102 | 512 | 0.100 | 0.01 | 5000 | 0.00001 |
| Pets | 8192 | 0.002 | 0.01 | 5000 | 0.00001 |
| Camelyon | 512 | 0.020 | 0.10 | 500 | 0.00100 |
| Resisc45 | 8192 | 0.020 | 0.01 | 500 | 0.00001 |
| sNORB-Azim | 24576 | 0.002 | 0.01 | 500 | 0.00001 |
| sNORB-Elev | 8192 | 0.050 | 0.01 | 500 | 0.00100 |
| Sun397 | 512 | 0.100 | 0.01 | 5000 | 0.00100 |
| SVHN | 24576 | 0.005 | 0.01 | 500 | 0.00001 |

Table 2: Hyper parameters selected for the VTAB-1k benchmark tasks when using pretrained ResNet-50 backbone. $F$: fraction of features kept, LR: learning rate, Steps: Training Steps.

## B    COST OF HEAD2TOE

We evaluate different values of $F$ and pick the value with best validation performance. Cost of HEAD2TOE consists of three parts: (1) $C_I$: Cost of calculating the representations using the pre-trained backbone. (2) $C_{all}$: cost of training the initial head $\boldsymbol{W}_{all}$ (in order to obtain $s_i$'s) (3) $\sum_f C_{F=f}$: total cost of validating different values of $F$. Cost of validating a fraction value $f$, assuming equal number of training steps, is equal to $C_f = C_{all} * f$. Therefore relative cost of searching for $F$ is equal to the sum of fractions validated (in comparison to the initial training of ($\boldsymbol{s}$)).

In Table 4, we compare cost of running HEAD2TOE adaptation with FINETUNING. HEAD2TOE uses the backbone once in order to calculate the representations and then trains the $\boldsymbol{W}_{all}$, whereas FINETUNING requires a forward pass on the backbone at each step. Therefore we calculate the cost

| Dataset | Target Size | F | LR | Steps | $\ell_{2,1}$ Coef. |
|---|---|---|---|---|---|
| Caltech101 | 768 | 0.050 | 0.01 | 5000 | 0.00100 |
| CIFAR-100 | 768 | 0.020 | 0.01 | 500 | 0.00001 |
| Clevr-Dist | 15360 | 0.002 | 0.01 | 500 | 0.00100 |
| Clevr-Count | 768 | 0.050 | 0.10 | 5000 | 0.00001 |
| Retinopathy | 768 | 0.010 | 0.01 | 500 | 0.00100 |
| DMLab | 32448 | 0.005 | 0.01 | 500 | 0.00001 |
| dSpr-Orient | 768 | 0.100 | 0.01 | 5000 | 0.00001 |
| dSpr-Loc | 32448 | 0.002 | 0.10 | 500 | 0.00100 |
| DTD | 768 | 0.100 | 0.01 | 500 | 0.00100 |
| EuroSAT | 768 | 0.100 | 0.01 | 500 | 0.00100 |
| KITTI-Dist | 32448 | 0.050 | 0.01 | 5000 | 0.00100 |
| Flowers102 | 768 | 0.020 | 0.01 | 500 | 0.00001 |
| Pets | 768 | 0.020 | 0.01 | 5000 | 0.00100 |
| Camelyon | 768 | 0.100 | 0.01 | 500 | 0.00100 |
| Resisc45 | 768 | 0.050 | 0.01 | 5000 | 0.00001 |
| sNORB-Azim | 32448 | 0.010 | 0.01 | 500 | 0.00001 |
| sNORB-Elev | 15360 | 0.200 | 0.01 | 500 | 0.00100 |
| Sun397 | 768 | 0.050 | 0.01 | 5000 | 0.00100 |
| SVHN | 32448 | 0.005 | 0.01 | 500 | 0.00001 |

Table 3: Hyper parameters selected for the VTAB-1k benchmark tasks when using pretrained ViT-B/16 backbone. $F$: fraction of features kept, LR: learning rate, Steps: Training Steps.

| Dataset | F | N | C | FLOPs (vs FINETUNING) | Size (vs FINETUNING) | Size (vs LINEAR) |
|---|---|---|---|---|---|---|
| Caltech101 | 0.010 | 467688 | 102 | 0.013932 | 0.020750 | 2.353167 |
| CIFAR-100 | 0.200 | 30440 | 100 | 0.002876 | 0.025743 | 2.977301 |
| Clevr-Dist | 0.001 | 467688 | 6 | 0.002808 | 0.000741 | 1.417419 |
| Clevr-Count | 0.005 | 30440 | 8 | 0.000270 | 0.000092 | 0.132278 |
| Retinopathy | 0.200 | 467688 | 5 | 0.002673 | 0.020531 | 47.099634 |
| DMLab | 0.020 | 467688 | 6 | 0.002808 | 0.003011 | 5.756287 |
| dSpr-Orient | 0.200 | 30440 | 16 | 0.002140 | 0.004183 | 3.001686 |
| dSpr-Loc | 0.005 | 467688 | 16 | 0.004154 | 0.002212 | 1.587624 |
| DTD | 0.005 | 1696552 | 47 | 0.023153 | 0.019157 | 4.692396 |
| EuroSAT | 0.100 | 30440 | 10 | 0.002088 | 0.001336 | 1.532776 |
| KITTI-Dist | 0.020 | 467688 | 4 | 0.002539 | 0.002215 | 6.350983 |
| Flowers102 | 0.100 | 30440 | 102 | 0.001094 | 0.013146 | 1.490882 |
| Pets | 0.002 | 467688 | 37 | 0.005181 | 0.002089 | 0.649417 |
| Camelyon | 0.020 | 30440 | 2 | 0.002018 | 0.000092 | 0.529114 |
| Resisc45 | 0.020 | 467688 | 45 | 0.008058 | 0.018474 | 4.725480 |
| sNORB-Azim | 0.002 | 1696552 | 18 | 0.010791 | 0.004851 | 3.094923 |
| sNORB-Elev | 0.050 | 467688 | 9 | 0.003212 | 0.009578 | 12.210897 |
| Sun397 | 0.100 | 30440 | 397 | 0.003678 | 0.049781 | 1.487498 |
| SVHN | 0.005 | 1696552 | 10 | 0.006884 | 0.005865 | 6.730334 |
| Average | | | | 0.005282 | 0.010729 | 5.674742 |

Table 4: Relative cost of HEAD2TOE when compared with FINETUNING and LINEAR. $F$ is the fraction of features kept, $N$ is the total number of features and $C$ is the number of classes. On average HEAD2TOE requires 0.5% of the FLOPs required by FINETUNING during the adaptation. Cost of storing each adapted model is also small: requiring only 1% of the FINETUNING and only 5.7x more than LINEAR. See main text for details on how the numbers are calculated.

of finetuning for $t$ steps as $C_I \cdot t$. Similarly, cost of HEAD2TOE is calculated as $C_I + C_{all} \cdot t$. The overall relative cost of $C_{all}$ increases with number of classes $C$ and number of features considered

$N$. As shown in Table 1-top, HEAD2TOE obtains better results than FINETUNING, yet it requires 0.5% of the FLOPs needed on average.

After adaptation, all methods require roughly same number of FLOPs for inference due to all methods using the same backbone. The cost of storing models for each task becomes important when the same pre-trained backbone is used for many different tasks. In Table 4 we compare the cost of storing different models found by different methods. A finetuned model requires all weights to be stored which has the same cost as storing the original network, whereas LINEAR and HEAD2TOE requires storing only the output head: $W_{linear}$. HEAD2TOE also requires to store the indices of the features selected using a bitmap. Even though HEAD2TOE considers many more features during adaptation, it selects a small subset of the features and thus requires a much smaller final classifier (on average 1% of the FINETUNING). Note that hyper parameters are selected to maximize accuracy, not the size of the final model. We expect to see greater savings with an efficiency oriented hyper-parameter selection.

## C  STANDARD DEVIATIONS FOR TABLE 1

In Table 5, we share the standard deviations for the median test accuracies presented in Table 1. On average we observe LINEAR obtains lower variation as expected, due to the limited adaptation and convex nature of the problem. Head2Toe seem to have similar (or less) variation then the other regularization baselines that use all features.

| | Natural | | | | | | | Specialized | | | | Structured | | | | | | | | Mean |
|---|---|---|---|---|---|---|---|---|---|---|---|---|---|---|---|---|---|---|---|---|
| | CIFAR-100 | Caltech101 | DTD | Flowers102 | Pets | SVHN | Sun397 | Camelyon | EuroSAT | Resisc45 | Retinopathy | Clevr-Count | Clevr-Dist | DMLab | KITTI-Dist | dSpr-Loc | dSpr-Ori | sNORB-Azim | sNORB-Elev | |
| Linear | 0.09 | 0.08 | 0.14 | 0.06 | 0.08 | 0.17 | 0.06 | 0.06 | 0.03 | 0.12 | 0.08 | 0.42 | 0.1 | 0.03 | 0.21 | 0.14 | 0.07 | 0.0 | 0.21 | 0.11 |
| +All-$\ell_2$ | 0.09 | 0.78 | 0.44 | 0.29 | 0.22 | 0.02 | 0.04 | 0.02 | 0.06 | 0.05 | 0.02 | 0.09 | 0.2 | 0.1 | 1.31 | 0.32 | 0.08 | 0.03 | 0.35 | 0.24 |
| +All-$\ell_1$ | 0.14 | 0.11 | 0.11 | 0.11 | 0.11 | 0.12 | 0.03 | 0.06 | 0.04 | 0.2 | 0.02 | 0.47 | 0.33 | 0.18 | 0.2 | 0.34 | 0.07 | 0.06 | 0.44 | 0.17 |
| +All-$\ell_{2,1}$ | 0.09 | 1.0 | 0.13 | 0.1 | 0.1 | 0.18 | 0.23 | 0.15 | 0.07 | 0.09 | 0.05 | 0.03 | 0.08 | 0.11 | 0.41 | 0.51 | 0.24 | 0.16 | 0.07 | 0.2 |
| Head2Toe | 0.14 | 0.25 | 0.08 | 0.08 | 0.24 | 0.24 | 0.16 | 0.23 | 0.06 | 0.06 | 0.03 | 0.18 | 0.23 | 0.13 | 0.43 | 0.3 | 0.06 | 0.4 | 0.08 | 0.18 |

| | Natural | | | | | | | Specialized | | | | Structured | | | | | | | | Mean |
|---|---|---|---|---|---|---|---|---|---|---|---|---|---|---|---|---|---|---|---|---|
| | CIFAR-100 | Caltech101 | DTD | Flowers102 | Pets | SVHN | Sun397 | Camelyon | EuroSAT | Resisc45 | Retinopathy | Clevr-Count | Clevr-Dist | DMLab | KITTI-Dist | dSpr-Loc | dSpr-Ori | sNORB-Azim | sNORB-Elev | |
| Linear | 0.1 | 0.23 | 0.19 | 0.16 | 0.06 | 0.06 | 0.08 | 0.09 | 0.08 | 0.06 | 0.0 | 0.06 | 0.02 | 0.07 | 0.92 | 0.21 | 0.08 | 0.08 | 0.03 | 0.14 |
| +All-$\ell_2$ | 0.13 | 0.17 | 0.0 | 0.66 | 0.36 | 0.04 | 0.56 | 0.02 | 0.59 | 0.01 | 0.04 | 0.23 | 0.13 | 0.12 | 1.24 | 0.28 | 0.09 | 0.84 | 0.2 | 0.3 |
| +All-$\ell_1$ | 0.06 | 0.11 | 0.15 | 0.08 | 0.19 | 0.31 | 0.08 | 0.08 | 0.06 | 0.12 | 0.05 | 0.28 | 0.17 | 0.23 | 0.87 | 0.34 | 0.07 | 0.22 | 0.28 | 0.2 |
| +All-$\ell_{2,1}$ | 1.55 | 0.03 | 0.12 | 0.04 | 0.06 | 0.41 | 0.13 | 0.18 | 0.08 | 0.1 | 0.04 | 0.09 | 0.13 | 0.15 | 0.87 | 0.66 | 0.1 | 0.23 | 0.22 | 0.27 |
| Head2Toe | 0.29 | 0.16 | 0.26 | 0.5 | 0.19 | 0.14 | 0.07 | 0.13 | 0.04 | 0.09 | 0.09 | 0.04 | 0.44 | 0.14 | 1.34 | 0.21 | 0.01 | 0.31 | 0.08 | 0.24 |

Table 5: Standard deviation of test accuracy over 3 seeds on the VTAB-1k benchmark using pretrained (**top**) ResNet-50 and (**bottom**) ViT-B/16 backbones. The mean column averages the standard deviations for each dataset.

## D  ADDITIONAL PLOTS FOR EXPERIMENTS USING SINGLE ADDITIONAL LAYER

Test accuracies when using a single additional intermedieate layer from a pretrained ResNet-50 backbone are shown in Fig. 6. *natural* datasets (except SVHN) are highly similar to upstream dataset (ImageNet-2012) and thus adding an extra intermediate layer doesn't improve performance much. However performance on OOD tasks (mostly of the tasks in *structured* category) improves significantly when intermediate representations are used, which correlates strongly with datasets in which HEAD2TOE exceeds FINETUNING performance.

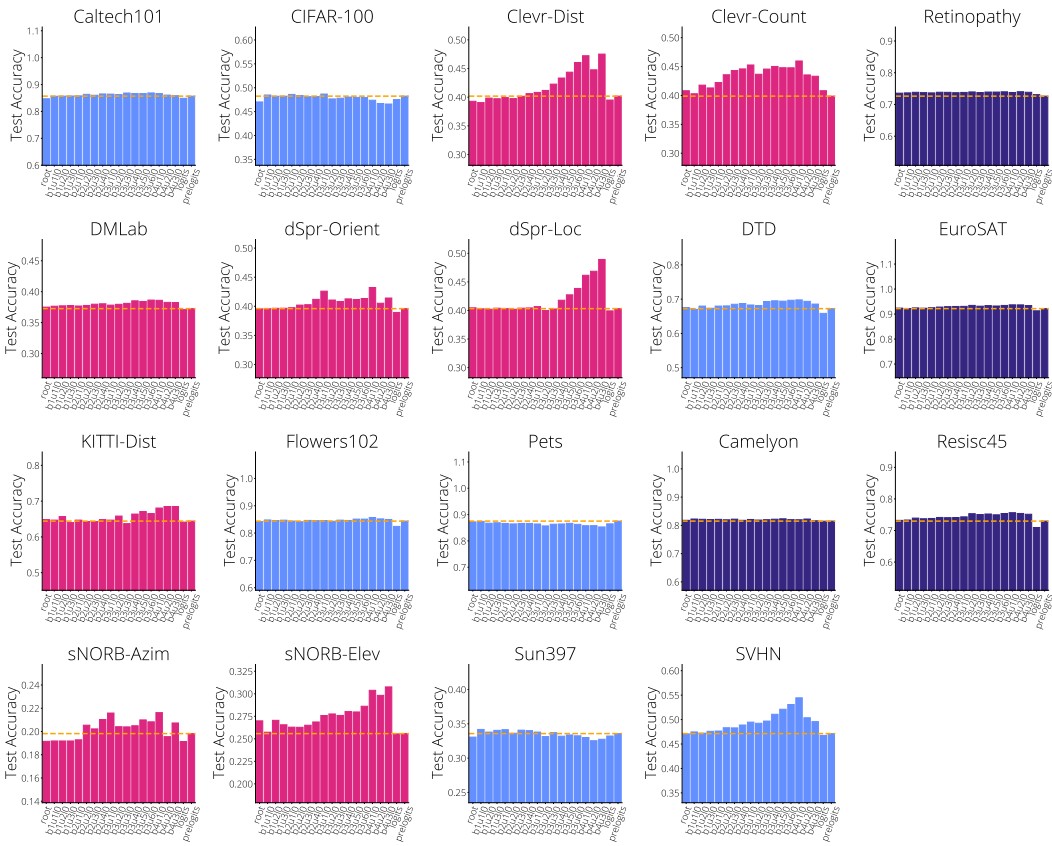

Figure 6: Test accuracies when using a single additional intermedieate layer from a pretrained ResNet-50 backbone.

# E    ADDITIONAL PLOTS FOR LAYER/FEATURE-WISE SELECTION COMPARISON

We compare layer-wise selection strategy discussed in Section 3 to HEAD2TOE in Fig. 7. For allmost all datasets, feature-wise selection produces better results. Retinopathy and Flowers-102 are the only two datasets where the layer-wise strategy performs better.

# F    ADDITIONAL PLOTS FOR RESNET-50 RESULTS

Improvement of HEAD2TOE over fine-tuning test accuracy for ResNet-50 backbone is shown in Fig. 8. Similar to earlier plots, we observe a clear trend between being OOD and improvement in accuracy: HEAD2TOE obtains superior few-shot generalization for most of OOD tasks. We also share the distribution of features selected for each task in Fig. 10. Since different tasks might have different number features selected, we normalize each plot such that bars add up to 1. Overall, features from later layers seem to be preferred more often. Early layers are preferred, especially for OOD tasks like Clevr and sNORB. We observe a diverse set of distributions for selected features, which reflects the importance of selecting features from multiple layers. Even when distributions match, HEAD2TOE can select different features from the same layer for two different tasks. Therefore, next, we compare the indices of selected features directly to measure the diversity of features selected across tasks and seeds.

**Transfer across Tasks**    In Fig. 9-left we investigate how features selected using a task $i$ performs when evaluated on a task $j$. Each pixel represents the average accuracy over 3 seeds. For each column we subtract the diagonal term (i.e. self-transfer, $i = j$) to obtain delta accuracy for each

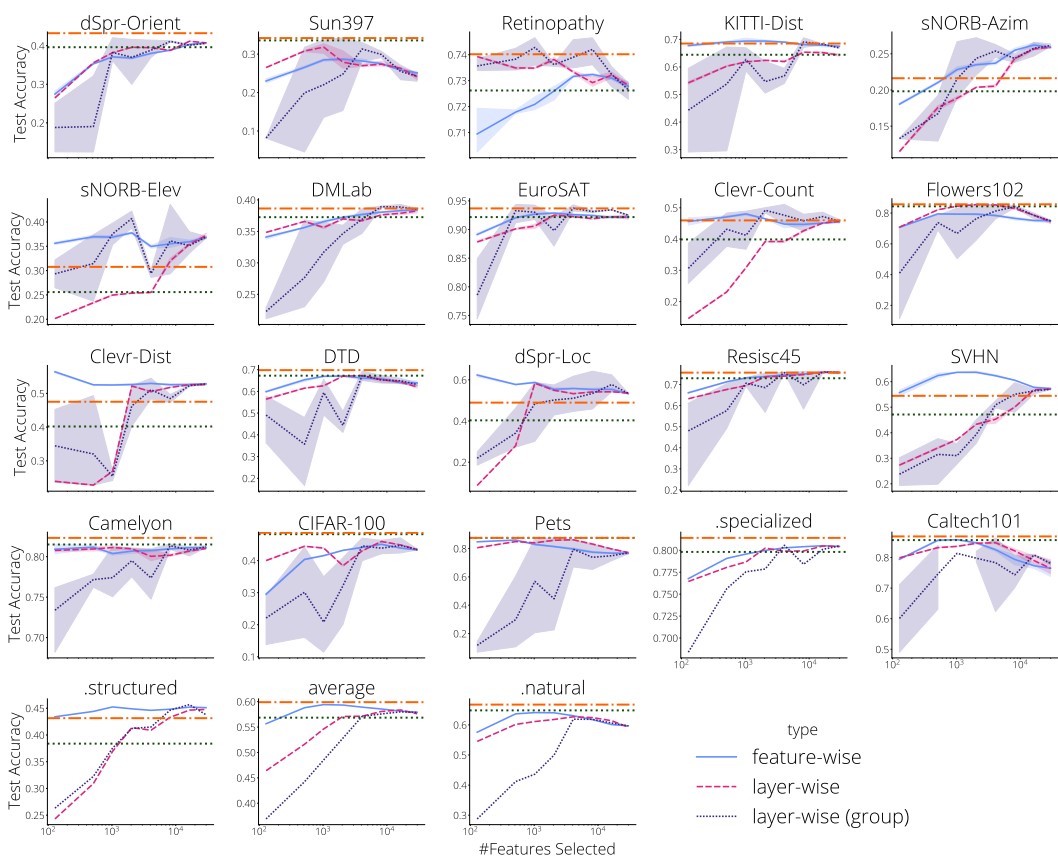

Figure 7: Test accuracies when varying the number of features selected for HEAD2TOE using a pretrained ResNet-50 backbone.

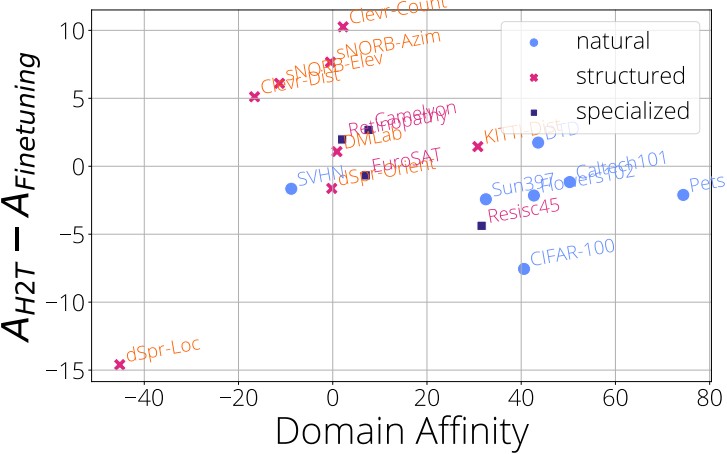

Figure 8: Accuracy improvement of HEAD2TOE compared to FINETUNING.

dataset. For most tasks, using a separate task for feature selection hurts performance. Results in Flowers-102 and Pets get better when other datasets like sNORB-Elev is used. Crucially no single dataset (rows of Fig. 9-left) seems to get best transfer, which highlights the importance of doing the feature selection during the adaptation (not beforehand). Note that, in practice (and in the literature) evaluation for each task is done in isolation, i.e. the data from other tasks are not available.

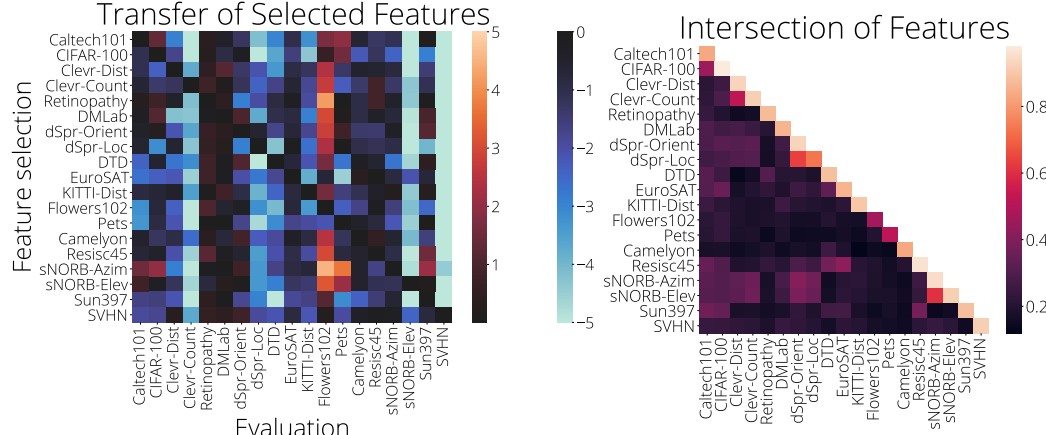

Figure 9: **(left)** Change in accuracy when features selected for a different task are used for adaptation. Most tasks get their best accuracy when the same task is also used for feature selection. **(right)** Intersection of features when selecting 2048 features from 29800 (same settings as in Fig. 4). The intersection is calculated as the fraction of features selected in two different runs. Values are averaged over 9 pairs of runs (3 seeds for each datasets), except the diagonal terms for which we remove identical pairs resulting in 6 pairs.

**Similarity of features selected**   In Fig. 9-right we investigate the intersection of features selected by HEAD2TOE. We select 2048 features for each task from the pool of 29800 features (same experiments as in Fig. 4-center). For each task HEAD2TOE selects a different subset of features. We calculate the fraction of features shared between each subset. For each target task we run 3 seeds resulting in 3 sets of features kept. When comparing the similarity across seeds (diagonal terms), we average over 6 possible combinations among these 3 sets. Similarly, when comparing two different datasets we average over 9 possible combinations. Results show that features selected by complementary tasks like dSprites-Loc and dSpr-Orient (or sNORB tasks) have high overlap. Both dSprites and sNORB images are generated artificially and have similar characteristics. This is reflected in the heat-map (about 40% overlap). Similarly, features selected for CIFAR-100 and Caltech101 tasks seem to have high overlap.

Features selected also vary across different runs (seeds). This variation seem to be higher for highly in-domain datasets like Flowers102 and Pets, which seems to correlate with HEAD2TOE's poor performance. Similarly dSprites-Loc seems to have higher variance across different seeds, which correlates with the outlier performance of HEAD2TOE on this task when compared with FINETUNING (see Fig. 8). We believe that understanding and reducing the inconsistency among different runs can be a promising direction to improve head-to-toe utilization performance.

Apart from a small fraction of datasets, most datasets seem to share less than 20% of the features, which highlights the importance of doing the feature selection for each target task separately.

**Effect of Training Data**   In Fig. 11, we compare the performance of HEAD2TOE with other baselines using reduced training data. Fraction ($d_f$)=1 indicates original tasks with 1000 training samples. For other fractions we calculate number of shots for each class as $int(1000 * f_d/C)$ where $C$ is the number of classes and then sample a new version of the task accordingly. For SUN-397 task, number of shots are capped at 1 and thus fractions below 0.75 lead to 1-shot tasks and thus results are all the same. Overall we observe the performance of HEAD2TOE improves with amount of data available, possibly due to the reduced noise in feature selection.

## G   ADDITIONAL PLOTS FOR SCALING BEHAVIOUR OF HEAD2TOE

Scaling behaviour of HEAD2TOE when using different feature target size and number of layers over 19 VTAB-1k tasks is shown in Fig. 12.

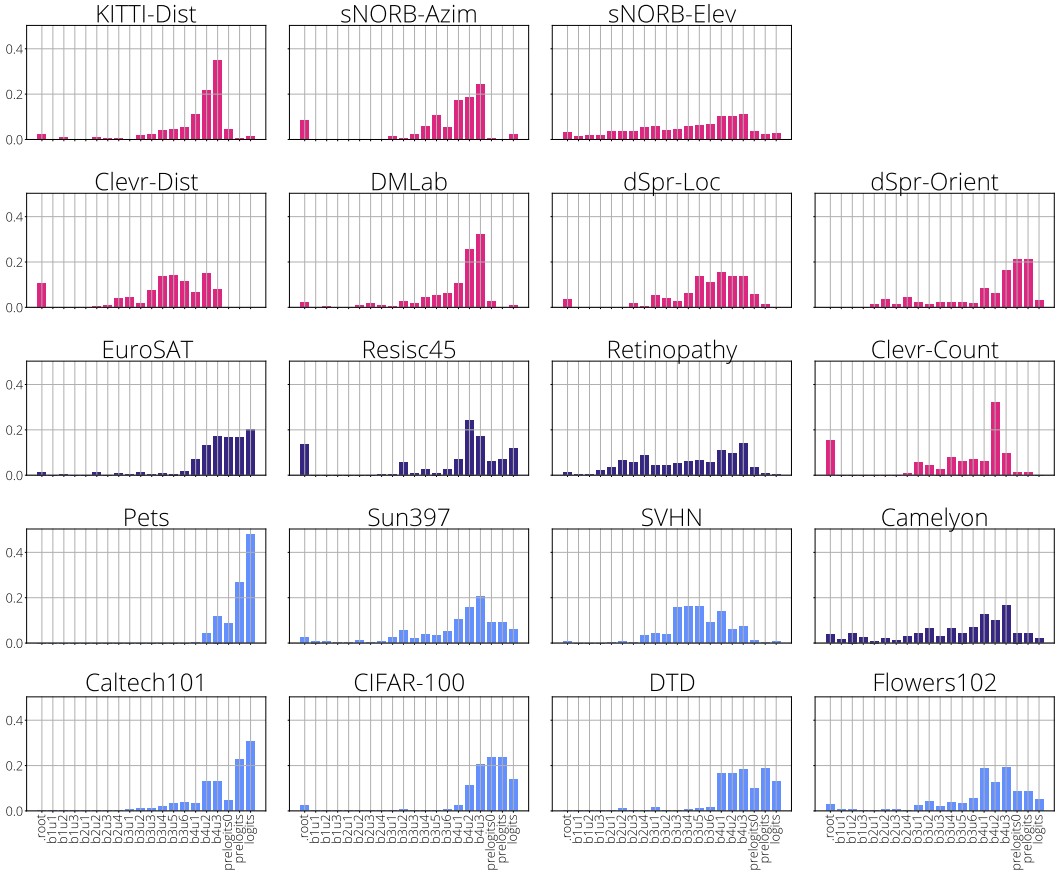

Figure 10: Distribution of selected features over different ResNet-50 layers for VTAB-1k tasks for results presented in Table 1. We group the layers in each block (group of 3 layers) to reduce numbers of bars.

## H  DETAILS OF INTERMEDIATE REPRESENTATIONS INCLUDED

**ResNet-50**   has 5 stages. First stage includes a single convolutional layer (root) followed by pooling. We include features after pooling. Remaining stages include 3,4,6 and 3 bottleneck units (v2) each. Bottleneck units start with normalization and activation and has 3 convolutional layers. We includes features right after the activation function is applied, resulting in 3 intermediate feature sets per unit. Output after 5 stages are average-pooled and passed to the output head. We include features before and after pooling. with the output of the final layer (logits), total number of locations where features are extracted makes 52.

**ViT-B/16**   models consists of multiple encoder units. Each attention head consists of a self attention module and followed by 2 MLPs. For each encoder unit, we include features (1) after layer-norm (but before self-attention) (2) features after self-attention (3-4) features after MLP layers (and after gelu activation function). Additionally we use patched image (i.e. tokenized image input), pre-logits and logits.

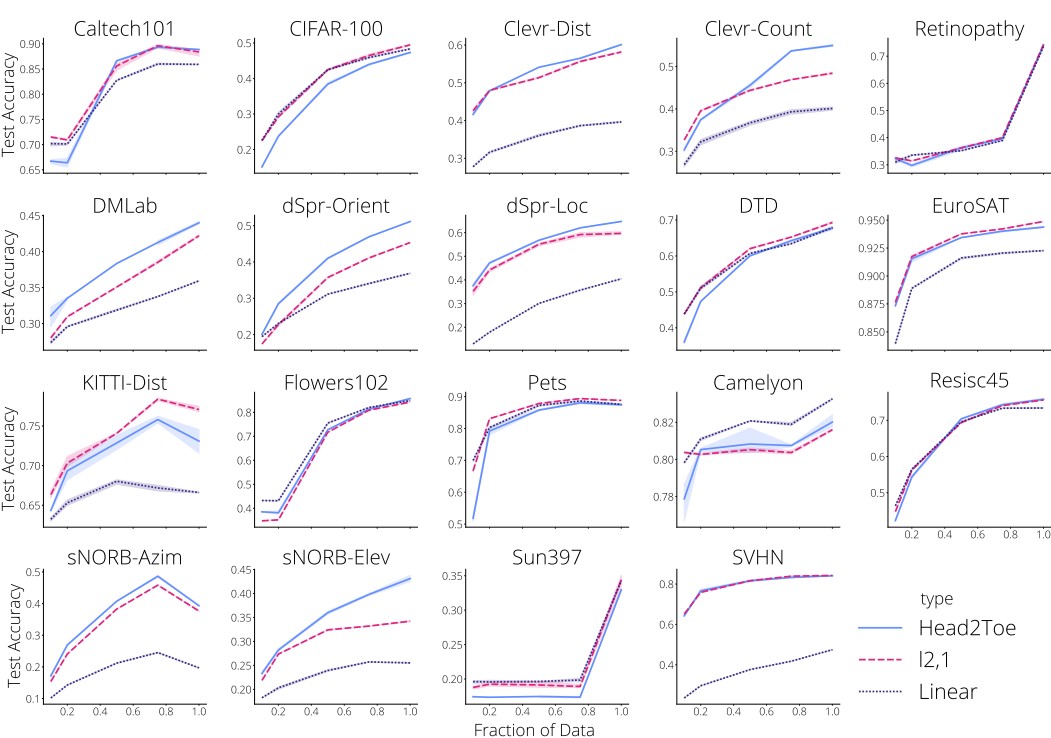

Figure 11: Effect of data available during training to the test accuracy. Fraction=1 indicates original tasks with 1000 training samples. Overall we observe the performance of HEAD2TOE improves with amount of data available, possibly due to the reduced noise in feature selection.

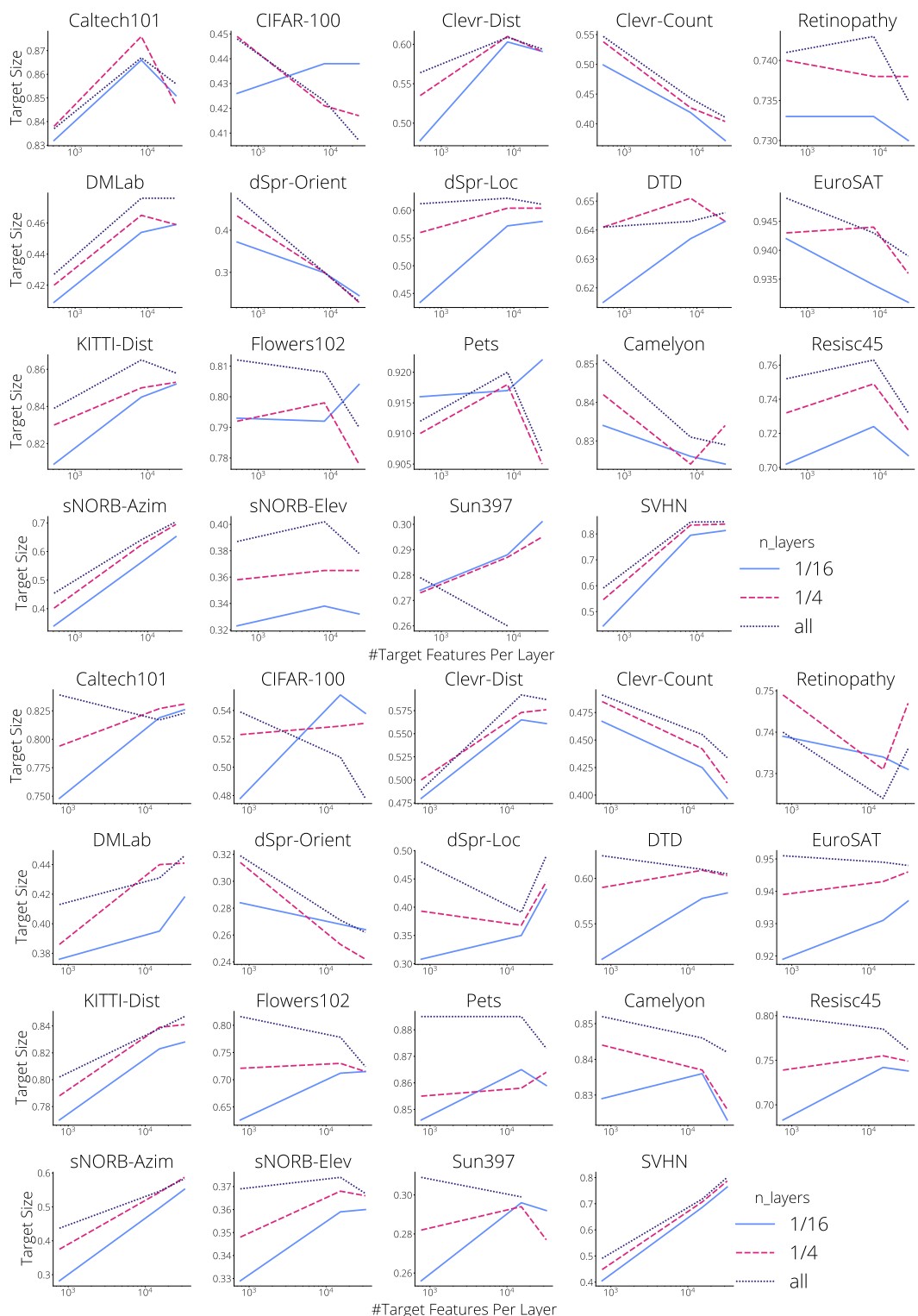

Figure 12: Scaling Behaviour over 19 VTAB-1k tasks when varying feature target size and number of layers utilized for **(top)** ResNet-50 and **(bottom)** ViT-B/16

