# OpenReview forum: "Head2Toe: Utilizing Intermediate Representations for Better OOD Generalization"
_ICLR.cc/2022/Conference — ICLR 2022 Submitted_

### Official Review · Reviewer_kz8L · 2021-10-29

**Correctness:** 3
**Technical Novelty And Significance:** 2
**Empirical Novelty And Significance:** 3
**Recommendation:** 6
**Confidence:** 3

**Main Review:**


# Evaluation

This is a well-written paper with a simple but clearly motivated idea. The preliminary experiments and discussion are helpful in motivating the approach. I am not familiar with the benchmark chosen for experimentation, but the experiments appear to be sound as far as I can assess. There is a comparison with several reasonable baselines.


# Questions and comments

1. While I'm not very familiar with the transfer learning literature in image classification, the idea seems common enough (as discussed in the related work) and pretty straightforward. This isn't necessarily a shortcoming, but I think it would make sense to discuss similar work in more detail, and perhaps draw more direct comparisons. Others have studied intermediate representations -- have they used the exact same approach? A different one? How would the results of the present approach compare to others that use intermediate representations? What exactly is the novelty?

2. The name Probing is not so intuitive. I'm familiar with this term from the interpretability literature, especially in NLP, where a probing classifier is used on frozen features to assess their quality w.r.t some property. In the present context, however, the goals are different, and a more familiar term (at least to me) is to refer to frozen transfer, or feature-based transfer learning as opposed to fine-tuning.

3. In figure 2, what is Scratch trained on? If it's a randomly-initialized ResNet-50 trained on low-resource datasets, it seems like a problematic case of over-parameterization. Would a simpler network perform better? the comparison with Linear is maybe problematic here.

4. The Control reported in section 3.1 is helpful in assessing the extent of improvement that can be expected from adding intermediate representations. Oracle is than 1% greater than Control on average.

5. The group Lasso regularization is applied once, and then a second final classifier is trained on the remaining features. I'm confused as to why this retraining is necessary, and why the group-lasso-regularized classifier can't be used as is. The experiments show that this second step works better, but why would it be so? Is it basically an optimization problem?

6. In group Lasso, it's important to a-priori select the groups. In one case, features are grouped by layer, but I'm confused about the "feature-wise strategy". What are the groups there? And, would it make sense to consider other groupings?

7. Related to the above, [1] performed feature selection with intermediate representations in an NLP context, and found different layers and different features inside a layer. They ahd a different kind of regularization, the ElasticNet. Even though their motivation is different (analyzing redundancy), they still get some pretty good results, so it would be interesting to compare the approaches.

8. In table 1, I would have liked to know which differences are statistically significant and which aren't. The text mentions standard deviations as appearing in Appendix 1, but I couldn't find them.

9. Could you discuss the different performances on the three groups (natural, structured specialized), of Head2Toe vs fine-tuning, in ResNet and ViT cases? Do the differences make sense in light of the nature of these categories?

10. From an interpretability standpoint, I would have liked to see more discussion of which features are selected, in which layers, and how this relates to different tasks and what we might expect.

[1] Dalvi et al., 2020. Analyzing Redundancy in Pretrained Transformer Models.


**Summary Of The Paper:**

# Summary

This paper proposes a method for using intermediate representations from a pre-trained model for better transfer to other tasks. A linear classifier is trained on features from multiple layers. A feature selection strategy is chosen, where first a linear classifier is trained on all features with group-lasso regularization, and then another classifier is trained only on features whose regularization score exceeds some threshold. This approach is evaluated in a low-resource transfer scenario of image classification, and compared both to training on all features without the two-step feature selection but with regularization, and to the standard fine-tuning approach. The proposed method works slightly better than fine-tuning with a ResNet model, and slightly worse with a ViT model, although the comparison is not completely fair, as discussed. There are different performances on different categories of images.



**Summary Of The Review:**

# Evaluation

This is a well-written paper with a simple but clearly motivated idea. The preliminary experiments and discussion are helpful in motivating the approach. I am not familiar with the benchmark chosen for experimentation, but the experiments appear to be sound as far as I can assess. There is a comparison with several reasonable baselines. See the main review for more questions and comments.

---

> ### Author Response · Authors · 2021-11-12
> **Reference Missing**
>
> We believe the reference [1] is missing in the feedback. It would be great if the reviewer can share the related work, so that we can include it in our response.
>
> Thanks

---

> > ### Comment · Reviewer_kz8L · 2021-11-15
> > **Added the missing ref**
> >
> > I've now added the missing reference in the official review. Sorry for this omission.
> > I'd be curious to hear how the proposed method is similar or different from [1]. The basic idea seems very similar, although [1] is in an NLP context and with different goals.

---

> > > ### Author Response · Authors · 2021-11-16
> > > **Comparison to [1]**
> > >
> > > Thanks for the reference. We were unaware of this highly relevant work from the NLP literature. We will add a comparison to [1] and to the earlier work of Dalvi et. al [2]. That being said, it is great to see similar ideas being explored in NLP, suggesting the potential impact of our findings to many different domains.
> > >
> > > Beyond the difference in choice of domain, our method and theirs differ in several critical respects:
> > >
> > > - **(Selecting features from all layers)** We use the output of *every layer* and *every token*. A self-attention block consists of a self-attention layer and 2 MLPs. For example, when utilizing the ViT model, we utilize both the input and output of the self-attention in addition to the output of the two MLP layers. So in total we probe 4 locations at every self-attention block instead of looking at the output only. In Fig. 5-right, we show the importance of including these extra locations. Furthermore, we use the representations of all tokens, whereas [1] uses only the features of CLS token. So their selection method consists of significantly fewer candidate features.
> > >
> > > - **(Feature selection method)** As the reviewer mentioned, the feature selection algorithms are different. We observed in our experiments group lasso works better than l1/l2 at feature selection.
> > >
> > > - **(Fine Tuning the Backbone)** We don't fine tune the backbone after feature selection whereas [1] fine tunes the entire backbone thus the cost of storing the adapted network is much larger than our approach. We expect our results to become even better with fine tuning the backbone. However, our key finding is that one can match fine tuning results using intermediate features without needing to fine tune the backbone.
> > >
> > > [2] https://arxiv.org/abs/1812.09355

---

> > > > ### Comment · Reviewer_kz8L · 2021-11-19
> > > > **Thank you for this comparison**
> > > >
> > > > Thank you for this good comparison. It'll be good to add it in the paper. Point 1 on features from all layers is especially interesting.
> > > > Minor remark: my reading of [1] is that they don't fine-tune the underlying BERT model. They use what they call a feature-based transfer learning, without fine-tuning the original model weights.

---

> > > > > ### Author Response · Authors · 2021-11-19
> > > > > **thanks for the response**
> > > > >
> > > > > We thank the reviewer for their response. On Section 3.1 of [1] (*Other Settings* paragraph) it says: `... CLS token's representation (from a fine-tuned model) is used for sequence classification`, which led us to believe that they do fine-tuning before the feature selection. We will clarify this with the authors and ensure that our comparison is correct and fair.

---

> ### Author Response · Authors · 2021-11-12
> **Author Response**
>
> We thank the reviewer for their thoughtful feedback. We hope our responses below addresses the individual questions and concerns raised. If so, it would be great to see this being reflected in the final recommendation.
>
> 1. **(Extended comparison to prior work)** Head2Toe fixes the backbone  and thus is better aligned with Linear in terms of both training time and storage cost (see the new Table:4 for the cost breakdown). Existing work on efficient fine-tuning (DiffPruning, Adapters) focuses on reducing storage cost and modifies the backbone itself. Head2Toe reduces the training cost significantly compared to these alternatives since it keeps the backbone fixed. Similarly, All of the previous work which uses intermediate layers focuses on selecting layers (i.e. which layers transfer best). In our work we show the importance of combining and selecting features from *all* intermediate layers, treating each feature independently, which is to our knowledge novel.
>
> 2. **(Probing)** We also struggled with terminology. As you mention, `linear probing` comes from the interpretability literature and used widely in the recent (https://arxiv.org/abs/2103.00020) and previous work (https://arxiv.org/abs/1910.04867); thus we decided to use it. Our notion of probing corresponds to the notion in the interpretability literature: using frozen features to assess–in our case–performance on a transfer task.
>
> 3. **(Scratch results)** In Fig. 2, all methods are trained on the VTAB-1k benchmark, where each task has 1000 training samples. We used Fine Tuning and Scratch results from the original VTAB paper. Overfitting may or may not be a problem (https://openai.com/blog/deep-double-descent/) for training from scratch. However, improving training from scratch is not our goal. We present Scratch results here to sort each task according to its domain affinity.
>
> 4. **(Control/Oracle experiments in section 3.1)** Results in Section 3.1 shows the extent of improvement that can be expected from adding ___a single___ intermediate layer. As shown in Fig. 5, adding more features from more layers improves performance of Head2Toe significantly leading to 65.8% average accuracy presented in Table 1.
>
> 5. **(whether retraining is necessary)** Group Lasso regularization can be applied with proximal updates to select features in one pass. However, then one needs to search for many regularization coefficients, which would require multiple passes on the full set of features, thus increasing the cost significantly. In contrast, Head2Toe does the initial costly training once and tries different thresholds to obtain best validation performance, which requires only 18% extra computation.
>
> 6. **(selection of groups)** Confusion over the term ‘group’ is completely understandable. The groups used in Head2Toe’s group lasso are the fan-out weights from individual features (see Equation 3 of main article). This grouping corresponds to `feature-wise` selection in Fig. 4. We also constructed a layer-wise alternative that still performs Lasso on the individual unit fan-out weights but then computes a layer-wise score by averaging the individual feature scores within a layer (`layer-wise`). In response to your comment, we have explored a third way of applying group lasso: to the vector of all fan-out weights from all units in a layer (see Fig. 4-center and Fig. 7) (`layer-wise-group`). The Head2Toe strategy beats the two strategies in which all units in a layer are selected in an all-or-none fashion.

---

> > ### Author Response · Authors · 2021-11-12
> > **Author Response (Cont.)**
> >
> > 8. **(standard deviations)** Thanks for noticing the missing tables. We added them to the Appendix (Table 5). On average, the standard deviation is around 0.2%.
> >
> > 9. **(different performances on the three groups)** Head2Toe fixes the backbone and is thus better aligned with Linear in terms of cost. In both the ViT and ResNet experiments, Head2Toe does significantly better than linear (+9-10%) and matches fine tuning with significantly less cost ( Table 4). Most of these gains are achieved when adapting to rendered artificial images with non-standard targets (_structured_) and specialized images using non-standard cameras (_specialized_), many of which are OOD compared to the upstream dataset (ImageNet). One of our main contributions is to demonstrate that intermediate representations are most useful in out-of-domain settings. Head2Toe does relatively better in the structured domain because of this, which is shown Fig. 2 (and Fig. 6). Further evidence is provided in Fig.10, which shows that Head2Toe improves most when the intermediate features are useful for the target task. Tasks in the natural-image category (_natural_) are most similar to the upstream dataset (Imagenet). Consequently,  using intermediate representations has  limited value for natural tasks.
> >
> > 10. **(additional results on interpretability)** We discussed and visualized the distribution of selected features by layer in Appendix Figure 8-right (initial submission). Following the feedback given, we extended these results in Appendix F. Fig. 10 shows the distribution of features selected by Head2Toe for each task. Overall, features from later layers seem to be preferred, but the distribution of selected features by layer varies from task to task. Even when distributions match, Head2Toe can select different specific features from a given layer for different tasks. Therefore, In Fig. 9-right, we compared the indices of selected features directly to measure the diversity of features selected across tasks and seeds. Apart from a small fraction of tasks, most task pairs share fewer than 20% of features in common, which highlights the importance of doing the feature selection for each target task separately. In Fig. 9-left we investigate how features selected using task $i$ performs when evaluated on a task $j$, which highlights similarities among different tasks. An extended discussion is provided in Appendix F.

---

> > > ### Comment · Reviewer_kz8L · 2021-11-15
> > > **Thanks for your response**
> > >
> > > Thank you for your response. It does help clarify some points I wasn't sure about. I'm still concerned about limited novelty, and defer to fellow reviewers about experimental issues in this domain. Therefore I will keep my current recommendation.

---

> > > > ### Author Response · Authors · 2021-11-16
> > > > **Thank you for your response**
> > > >
> > > > We believe the novelty of a work should include the novelty of its findings as discussed in the [reviewer guide](https://iclr.cc/Conferences/2021/ReviewerGuide). In addition to the algorithmic novelty, our work presents the following experimental findings:
> > > >
> > > > - Using intermediate features improves OOD transfer *beyond* SOTA (i.e., fine tuning) without fine tuning the backbone itself.
> > > > - Including more activations/layers in the initial pool during feature selection monotonically improves transfer performance.
> > > >
> > > > Given the difficulty of doing transfer learning on OOD tasks and the ever growing network sizes, these novel findings have a non-trivial potential to inform future research.
> > > >
> > > > We thank the reviewer for their reply.

---

### Official Review · Reviewer_VhK8 · 2021-11-03

**Correctness:** 3
**Technical Novelty And Significance:** 3
**Empirical Novelty And Significance:** 3
**Recommendation:** 5
**Confidence:** 3

**Main Review:**

Overall, an interesting paper. The proposal is indeed interesting, although utilizing intermediate representations has been seen in the past.
My main concern regarding authors claim "We conjecture that FINETUNING
better leverages existing internal representations rather than discovering entirely new representations; FINETUNING exposes existing features buried deep in the net for use by the classifier". Even if Head2Toe manages to reach comparable performance to finetunning, that doesn't proves that existing good representations where present in intermediate layers and where "brought up".  An analysis on the resulting weights of both methods might be a better way to prove that.
In addion, Table 1 shows a gap between the best performance reached by Head2toe and the best performance using finetunning. That result seems to imply that is more than a clever usage of existing representations what finetunning the whole model is doing.


**Summary Of The Paper:**

The main focus of this paper is the study the use of intermediate layers in a deep pretrained model on downstream tasks.
Authors argue that the the fact that the good performance while fine tuning on downstream tasks even if data scarce is due to the prior existence of useful representations deep in the model which are brought up during training.
Authors propose a new approach. Head2toe, which consist in utilizing intermediate representations, while freezing the weights of the network for training for downstream tasks
Experimentation is carried out starting with a ResNet-50 and ViT-B/16 models on a variety of datasets collected on the VTAB collection, showing that the approach where Head2toe  outperforms linear finetunning (training a classification head on top of the model, while freezing the rest of the weights) and matches the performance of finetunning the whole model.

**Summary Of The Review:**

An interesting idea,
Numbers hold
Claims are not completely justified with experimentation.

---

> ### Author Response · Authors · 2021-11-12
> **Author Response**
>
> We would like to thank the reviewer for raising their concern. We agree our results don't prove the motivation presented in Section 2, we have updated the last paragraph of Section 2 to reflect this point. Our conjecture comes from the observation that fine tuning must use (and possibly transform) some information from intermediate features, or else it would be no better than linear probing or training from scratch. We use this as a motivation for probing intermediate features; however, showing how this is achieved is beyond the scope of our work.

---

### Official Review · Reviewer_nxiA · 2021-11-03

**Correctness:** 2
**Technical Novelty And Significance:** 2
**Empirical Novelty And Significance:** 2
**Recommendation:** 5
**Confidence:** 4

**Details Of Ethics Concerns:**

I think there is not any ethics concerns.

**Main Review:**

Strengths:
1. This paper investigates how to use pre-trained models for better transfer learning, which is an important problem.
2. The paper is generally well-written and easy to follow.
3. The proposed method is simple, technically sound, and makes sense.
4. The authors conducted extensive experiments on VTAB benchmark with both ResNet and ViT backbones, and conducted comparisons with several baselines. The overall performance of Head2Toe shows some improvement compared to linear prob and even fine-tuning.
5. The preliminary experiments and analysis is informative and helps the readers understand to motivation of the paper.

Weaknesses:
1. The idea of combining intermediate representation from a pre-trained model to improve fine-tuning (transfer) performance is already exploited by ELMo. They use a learnable weight to control the relative importance of features from different layers. The key difference between Head2Toe and ELMo's approach is that Head2Toe selects a subset of intermediate features after learning the weights. However, I'm not sure this makes the novelty, or technical contribution, enough.
2. The experimental results is not convincing enough. In the VTAB benchmark, in natural and structured category, the performance of Head2Toe is only comparable with linear prob but worse than fine-tuning. The overall improvement of Head2Toe is because it performs well on Specialized category and it has more tasks. It will be better if some analysis on why Head2Toe perform particularly on Specialized category but not on others.
3. The claim in the title and abstract that Head2Toe improve OOD generalization are not well supported by the experimental results. The main experiments only show that Head2Toe improves transfer performance. However, the fine-tuned models' performance on OOD samples is not tested.

**Summary Of The Paper:**

This paper propose Head2Toe, a method that exploits intermediate representations of DNNs to improve performance, and OOD generalization of transfer learning. The key idea of the Head2Toe is to augment traditional linear probing with intermediate representations. It uses group lasso as regularization to learn weights of different features, and then select features based on the learned weights and validation performance.Experiments on VTAB benchmark show that Head2Toe outperforms linear probes and is competitive compared with fine-tuning. An important finding is that Head2Toe improves OOD generalization.

**Summary Of The Review:**

The paper is generally well-written and the method makes sense. However, given that the novelty of the method and some concerns about experimental results, I think the paper, in its current form, is slightly below the bar. I believe this paper can be improved largely if theconcerns about experiments are resolved.

---

> ### Author Response · Authors · 2021-11-12
> **Author Response**
>
> We thank the reviewer for their feedback and for highlighting the submission's strengths. Below we respond to specific questions.
>
> 1. **(Similarity to ELMo)** We thank the reviewer for making us aware of this important work. Head2Toe and ELMo use intermediate features differently and we have added a discussion on this point to the related work section: "ELMo averages intermediate representations of a language model (2 LSTM embeddings + the token embedding) using a learned linear combination (a softmax). ELMo requires embeddings to be the same size and is most similar to the suboptimal layer-wise selection baseline shown in Fig. 4-center." Intermediate representations used in our work vary in dimensionality; therefore, ELMo is not applicable to our setting. Head2Toe doesn't average layer representations; instead it selects a subset of features from many layers and trains a linear classifier on the concatenation of the selected features. We also highlight that one of our core contributions is to relate the benefit of selecting intermediate representations with the OOD nature of the downstream image-classification task. We hope this clarifies the reviewer's novelty concerns.
>
> 2. **(discussion of results)** Head2Toe fixes the backbone and is thus is better aligned with Linear in terms of cost (see the new Table 4 for a cost breakdown). In both the ViT and ResNet experiments, Head2Toe does significantly better than linear (+9-10%) and matches fine tuning with significantly less cost (See Table 4). As the reviewer pointed out, most gains are achieved in structured domain tasks, many of which are OOD compared to source dataset (ImageNet). One of our main contributions is to demonstrate that intermediate representations are most useful in out-of-domain settings. Head2Toe does relatively better in the structured domain because of this, which is shown Fig. 2 (and Fig. 6). Further evidence is provided in Fig.10, which shows that Head2Toe improves most when the intermediate features are more useful for the target task.
>
> 3. **(improving OOD generalization)** OOD is an overloaded term with no universally agreed-upon definition. In our work, OOD-ness is based on our definition of domain affinity, which is a legitimate definition, if arguably one among a number of other definitions. We use OOD in our work in the context of transfer learning and measure OOD transfer/adaptation performance, demonstrating the benefit of using intermediate features when adapting to OOD tasks. If the reviewer can provide a specific alternative definition, we are happy to consider that in our experiments.

---

> ### Author Response · Authors · 2021-11-28
> **Remaining Concerns**
>
> Approaching to the end of the discussion period, we like to thank the reviewer again for their time and feedback. Did our response address reviewers' concern on novelty and experiments? As explained our response, our results show the importance of selecting individual features (compared to averaging them like done in ELMo) and we hope the reviewer can find the time to look at our response. If their concerns are all addressed would the reviewer consider raising their scores?

---

> > ### Comment · Reviewer_nxiA · 2021-11-30
> > **Response to author feedbacks**
> >
> > I have read the authors' feedback. It addresses some of my concerns. However, I'm still conservative about the technical contribution upon ELMo's method. I believe the proposed method is equivalent to adding sparse constraints on ELMo's weights and make the weights in neuron level. Therefore I am not convinced enough to raise the score, but it will not upset me to see this paper being accepted.

---

> > > ### Author Response · Authors · 2021-11-30
> > > **Novelty Concerns**
> > >
> > > Reviewer says that: "the proposed method is equivalent to adding sparse constraints on ELMo's weights and make the weights in neuron level.". Even with those changes (which are quite significant), ELMo can't handle representations with different dimensions. Furthermore novelty of a paper should include novel experimental results and observations. Apart from the method itself, we demonstrate (1) the a strong correlation between good OOD domain adaptation and using intermediate features (2) positive scaling when more the layers and features included in the feature selection.
> > >
> > > Regardless, we thank the reviewer for taking the time to read our response and their valuable feedback.

---

### Official Review · Reviewer_yqBZ · 2021-11-03

**Correctness:** 3
**Technical Novelty And Significance:** 3
**Empirical Novelty And Significance:** 3
**Recommendation:** 6
**Confidence:** 5

**Main Review:**

Strengths:
- The paper is very well-written and easy to follow.
- The overall idea of Head2Toe for transfer learning is interesting.
- Competitive performance on VTAB benchmark.

Weaknesses: While the main idea and experiments presented in the paper are very interesting, there are several important weaknesses in the paper that need to be throughly addressed to improve the quality of the work (mainly missing experiments, analysis and comparisons).

- While authors empirically demonstrate the utility of intermediate features in experiments, there are no feature visualizations to support the claims. How about visualizing the selected intermediate features per target task? Is it possible to analyze the weight matrix W across different target datasets to better understand what type of features being selected for transfer learning? Are the selected features complementary go each other?

- In current experiments, features are selected specific to one target task. How about selecting a common set of intermediate features for all the tasks? How is that global baseline comparable to individual selecting selection of features per task?

- I would suggest the authors to rearrange or rewrite the main contributions of the paper. Points 1-5 in introduction section are not really the contributions. In other words, authors should merge them into 2-3 key contributions along the directions such as, problem definition, methodology and experiments.

- How the proposed method comparable to finetuning on other tasks besides image classification, e.g., video classification? I believe the main conclusions will still be the same but a small experiment on this will strengthen the paper.

- Authors compare with layer wise selection by computing the mean relevance score of each feature in a layer. Can it be not done through the L21 optimization directly where group of features belong to one layer will be made zero?

- Authors mention that one of the properties of the proposed method is key for transfer-learning methods to be practical with billions of parameters such as GPT-3. I wonder how is the proposed method applicable to transfer learning in NLP. Experiments and analysis on text classification or other standard NLP tasks should be included in the paper to validate the effectiveness of the proposed method beyond image classification.

- Is it possible to include some of the tweaks used in ViT paper to further improve the performance of Head2Toe (say Head2Toe+), e.g., different aggregation schemes as authors mentioned in the main paper?

- How is the proposed method comparable to Linear probing and Finetuning with different percentages of labeled data in the target domain? Is it still effective in low data regime, e.g., 1% of labeled samples?

- How is this method comparable to existing transfer learning methods besides Finetuning on VTAB benchmark, e.g., selective finetuning or some of the methods mentioned by the authors in the related work?


**Summary Of The Paper:**

This paper explores the utility of intermediate layers for linear probing in transfer learning. Specifically, authors proposed Head-to-Toe
probing (HEAD2TOE), that selects features from all layers of the source model to train a classification head for the target-domain. Experiments on the VTAB benchmark shows that Head2Toe matches performance obtained with fine-tuning on average, but critically, for out-of-distribution transfer, Head2Toe outperforms fine-tuning.

**Summary Of The Review:**

Given the interesting idea and competitive performance, my initial recommendation is borderline accept. I would like to see authors response on new experiments and discussions during the rebuttal as I think the experiments are limited and somewhat unconvincing in the current version of the paper.

---

> ### Author Response · Authors · 2021-11-12
> **Author Response**
>
> We thank the reviewer for their valuable suggestions. We are happy to see our work triggers many follow-up questions and experiments. We have expanded our analyses and discussion to the extent possible within page limits. We hope this addresses concerns on experimental evidence. We respond to specific questions and suggestions below.
>
> - **(feature visualizations to support the claims)** We discussed and visualized the distribution of selected features by layer in Appendix Figure 8-right (initial submission). Following your suggestions, we extended these results in Appendix F. Fig. 10 shows the distribution of features selected by Head2Toe for each task. Overall, features from later layers seem to be preferred, but the distribution of selected features by layer varies from task to task. Even when distributions match,  Head2Toe can select different specific features from a given layer for different tasks. Therefore, In Fig. 9-right, we compared the indices of selected features directly to measure the diversity of features selected across tasks and seeds. Apart from a small fraction of tasks, most task pairs share fewer than 20% of features in common, which highlights the importance of doing the feature selection for each target task separately. An extended discussion is provided in Appendix F.
>
> - **(selecting a common set of intermediate features)** Results presented in Fig.6 show that different tasks prefer features from different layers. Similarly Fig. 9-right shows that features that achieve the best results are relatively disjoint across tasks. Therefore, we expect that using a common set of features would achieve lower performance. This prediction is confirmed in Fig. 9-left, where we investigate how features selected using task $i$ perform when evaluated on a task $j$. Each pixel represents the average accuracy over 3 seeds. For each column we subtract the diagonal term (i.e. self-transfer, $i=j$) to obtain delta accuracy for each dataset used in feature selection. For most tasks, using another task for feature selection hurts performance (exceptions are Flowers-102 and Pets). Crucially, no single dataset (rows of Fig. 9-left) seems to find the best set of features, which highlights the importance of doing the feature selection during the adaptation (not beforehand). We also note that, in practice (and in the literature), evaluation for each task is done in isolation, i.e. the training set of other tasks are not available.
>
> - **(main contributions of the paper)** We incorporated the feedback given and merged our contributions into 3 main points. The reviewer may think of "contributions" as novel methods and state-of-the-art results, but we wish to emphasize that we consider our contributions to include a deeper understanding of transfer learning which helps to motivate Head2Toe. Perhaps as important as the method itself are our observations that (1) intermediate features are most useful when transferring to OOD tasks and (2) increasing the size of the potential feature pool leads to a better performance (perhaps surprising for few-shot adaptation).
>
> - **(L21-norm on layer-wise groups)** We did run experiments using layer-grouped $\ell_{2,1}$ norm, but omitted the results due to its subpar performance. We added these results back to Fig4-center (it follows a similar trend as the averaged version) and similarly updated the rest of the plots (Appendix Fig. 7).
>
> - **(benchmarking in video classification and nlp)** Papers that benchmark across various modalities are rare, possibly due to space constraints and differences in evaluation. In this work we focus on image classification. We look forward to applying Head2Toe to other domains, especially NLP. We share the reviewer's confidence that our conclusions will carry across domains. We believe an in-depth study of Head2Toe in a different domain would require different architectures, training methods and baselines and thus is better studied in a future work.
>
> - **(Head2Toe+ for ViT)** As mentioned in the main text, adding gradient clipping to finetuning improves results by 3%. We applied gradient clipping to Head2Toe, but since our optimization includes a single layer, this didn't affect the results. We will mention this in the paper.
>
> - **(different percentages of labeled data)** The VTAB-1k benchmark consists of 1000 samples per domain and a varying number of classes (Table 4). Consequently, the number of samples per class varies across tasks. Following your suggestions, we added Fig. 11 in which we reduce the number of samples available for each task further and compare Head2Toe with other baselines. Fraction=1 corresponds to the original experiments with 1000 training samples. Overall we observe the ordering among methods stay same, however the gap between Head2Toe and others narrow down with less data, possibly due to the increased noise in feature selection.

---

> > ### Author Response · Authors · 2021-11-12
> > **Author Response - Cont.**
> >
> > - **(comparison to existing transfer learning methods)** As mentioned above, our work focuses on improving linear probing with a minimal increase in cost, and more generally, our work aims to demonstrate the effectiveness of using intermediate features directly. Methods like DiffPruning or adapters require a forward pass at every training iteration and thus are much more costly to adapt (e.g., they can't be used easily when fine tuning GPT-3).

---

> ### Author Response · Authors · 2021-11-28
> **End of the Discussion Period**
>
> Given the end of the discussion period is tomorrow, we like to thank again the reviewer for their detailed review and helping us improving our paper. Did our response address reviewers' concerns? We added multiple sections and experiments (figures), which includes: (1) the diversity of features selected (2) suboptimal performance of selecting a single set of features (3) experiments with layer-grouped norm (4) performance on different data-regimes. We hope the reviewer can find the time to look at them. If the reviewer is satisfied with the new results and their concerns are addressed, would they consider raising their score and support acceptance? We thank again the reviewer for their time.

---

> > ### Comment · Reviewer_yqBZ · 2021-11-30
> > **Comments after reading author's rebuttal**
> >
> > I apologize for the delay. I sincerely thank the authors for the rebuttal and the revised draft. After carefully reading the author's rebuttal (and also other reviewer's comments), I am keeping my original rating of marginal acceptance but will not strongly argue for final acceptance. While I feel the paper has potential and the idea is interesting, the experiments in the current version of the paper have few limitations. The rebuttal addressed some of my concerns. However, I am not satisfied with the response on selecting a common set of intermediate features and benchmarking in video classification. First, benchmarking on classification tasks from a different domain (e.g., video or nlp) is essential to verify the effectiveness of the proposed method as the method is focus on linear/finetuning evaluation which is general in transfer learning. Second, comparison and analysis with a baseline that focus on selecting a common set of intermediate features should also be included. Authors response on selecting a common set of intermediate features only focuses on transferring features selected for one task to another (not the common set of features selected through optimization). Finally, authors mention about limitation of prior methods in terms of more costly to adapt (e.g., they can't be used easily when fine tuning GPT-3) but did not provide any supporting experiments to demonstrate their advantages in this case or in the experiments related to showing effectiveness on other tasks.

---

> > > ### Author Response · Authors · 2021-11-30
> > > **Thank you for your response**
> > >
> > > We thank the reviewer for reading our response and sharing their feedback. Reviewer indicates 2 limitations:
> > >
> > > (1) *Not having experiments on NLP and video classification* We believe this shouldn't be a requirenment for any paper to be accepted. We run experiments on 19 different image classification experiments, which is more than many papers published in the area.
> > >
> > > (2) *selecting a common set of intermediate features through optimization* Reviewer proposes a new idea for selecting common set of features (instead of selecting them for each task separately). Such algorithms doesn't exist in the literature as far as we are aware and coming up with a novel idea/method to do is clearly out of our scope. We also like to remind that each task needs to evaluated independently and one can only use upstream data (Imagenet) to select common set of features. However given the failure of linear probe for many of the OOD tasks, we expect this approach to not work well. We can add this baseline in the next version.
> > >
> > > Finally, we provide evidence for Head2Toe being less costly than finetuning based alternatives. Table-4 shows that on average Head2Toe requires 0.5% of the FLOPs needed for Finetuning (also the alternatives).

---

> > > > ### Comment · Reviewer_yqBZ · 2021-11-30
> > > > **Followup Response**
> > > >
> > > > Thanks for the quick response. Let me clarify the three main points/limitations (not two) that I made in my last comments.
> > > >
> > > > - First, I fully understand that experiments on NLP and video classification are not a requirement for the acceptance. However, experimenting on a couple of small video datasets by considering an already pretrained model (publicly available) will improve the experiments of the current paper. The title of the paper says that *Utilizing Intermediate Representations for Better OOD Generalization* which I thin should not be limited to only image classification datasets for a good paper.
> > > > - Second, for a common set of features: can we not process each task individually and then choose a common set of features? This is a baseline which I think can be easily compared to see how the proposed method is task-adaptive. This may not be practical but I don’t think the current paper anywhere talks about the method's practical applications.
> > > > - Third, authors claim about the disadvantage of prior methods on finetuning GPT-3 both in the main paper (submitted version) and in the rebuttal. The related statement in the submitted version is *Such properties are key for transfer-learning methods to be practical when using models with billions of parameters such as GPT-3 (Brown et al., 2020)* and in rebuttal is *Methods like DiffPruning or adapters require a forward pass at every training iteration and thus are much more costly to adapt (e.g., they can't be used easily when fine tuning GPT-3)*. Surprisingly, I don't find this statement related to GPT-3 and the reference in the revised version anymore (Did authors remove this in the revised version?). I would like to see experiments to verify this. More importantly, if experiments on other domains like NLP is not a requirement for acceptance (as authors say), then I think such statements or claims should not be made by the authors in the paper.
> > > >
> > > > To summarize, although I liked the overall idea of the paper, the paper has significant limitations that need to be propely addressed. Thus, I am keeping my initial score and will not argue for acceptance in ICLR.

---

> > > > > ### Author Response · Authors · 2021-11-30
> > > > > **Response**
> > > > >
> > > > > (1) Does the reviewer has a specific setting in mind for video classification? We can evaluate our algorithm before camera-ready using our backbones.
> > > > >
> > > > > (2) Using other evaluation tasks wouldn't be fair. As mentioned before, in transfer learning, each task is evaluated in isolation and a second task can't be used to improve performance (or select features). Furthermore it's not straight-forward how one would select common features after selecting features for each task individually (we are talking about a new method here). Voting or averaging might work, but maybe one should meta-learn such representations? We believe investigating these questions would be an interesting research project and we can mention this in our discussion.
> > > > >
> > > > > (Practicality) our method has a very similar cost as the linear probe, which is probably the most practical and efficient transfer learning method. We like to point out the Table-4, in which we show how our method compare in terms of FLOPs and parameter count.
> > > > >
> > > > > (3) Large models like GPT-3 or ViT-H are trained on hundreds of accelerators, thus they need a similar amount of accelerators to transfer to a new task. We removed reference to GPT-3, as our point is not specific to any domain or a model. The rest of the sentence stays as it is in the last paragraph of Section 3 which we share here as a reference:
> > > > >
> > > > > > Note that HEAD2TOE’s use of a fixed backbone means that as we search for features to include, the extracted features themselves are not changing. Consequently, we can calculate them once and reuse as necessary, instead of recalculating at every step, as required for FINETUNING. Furthermore, since the backbone is frozen, the only additional weight storage required for each target task is the output head. Such properties are key for transfer-learning methods to be practical when using models with billions of parameters.
> > > > >
> > > > > We provide evidence for this paragraph in Appendix B (Table-4). Methods like adapters and Diffpruning, has same (or more) cost as the finetuning, however they reduce storage cost of the adapted model similar to Head2Toe. We can add cost of these methods easily to Table-4.
> > > > >
> > > > > Finally we like to point out that out of 6 different experiments suggested by the reviewer we performed 4 of them (sparsity distributions, layer-grouped regularisation, effect of data available, feature visualization) We sincerely believe those experiments improved our work and we thank the reviewer for that.

---

> > > > > > ### Comment · Reviewer_yqBZ · 2021-11-30
> > > > > > **Final Comments**
> > > > > >
> > > > > > I sincerely thank the authors for their efforts in the rebuttal and also on engaging with reviewers to clarify the concerns. (1) I think any transfer learning experiment besides image classification will improve the current paper in justifying the author's claim on their method is not specific to any domain or a model (as also mentioned in the last response from the authors). One such experiment could be evaluating Head2Toe on video classification, e.g., evaluating performance of Kinetics400 pretrained model on UCF101 and HMDB datasets. (2) *Selecting common features is not a new method.* This is in fact a simple baseline that could be easily compared and analyzed by using majority voting after selecting features for each task individually. This will potentially help to understand task-adaptive nature of the proposed method. (3) Since authors are not experimenting with GPT-3 or ViT-H, I would recommend authors to not make any statements related to that in the paper.

---

> > > > > > > ### Author Response · Authors · 2021-12-01
> > > > > > > **Final Response**
> > > > > > >
> > > > > > > (1) We agree that it would be a strength to show results in different domains, but given 9 page limit; it normal for conference papers to demonstrate improvements on a single domain. Adapters and DiffPruning papers only study NLP transfer, many other papers also only study image classification. Again we have results on 19 tasks and we use architectures representing the two most popular architecture families: resnet and transformers. We will work on running video classification experiments, we thank the reviewer for the pointers.
> > > > > > >
> > > > > > > (2) As explained it can't be a simple baseline, since there is only one single upstream dataset (i.e. imagenet) to make a common feature selection and it is unlikely to improve performance as explained above. We can add this baseline easily if needed. We can also have a non-standard evaluation using all datasets. However, given the picture in Fig-9left, where we observe negative transfer between task features, this would also be unlikely to improve performance, but we can still add those baseline if needed.
> > > > > > >
> > > > > > > (3) Training difficulty of GPT-3 and ViT-H can be given by numbers (i.e. training flops, CO2, number of accelerators, time) and one doesn't need to run experiments to argue the cost of it. Regardless we don't mention these models in the latest version of the paper, so this concern is hopefully addressed. All we intended to do say is (1) Finetuning is costly, using intermediate features can reduce cost significantly and bring better accuracy than linear probe. We provide numbers to back these claims up (Appendix B).

---

### Decision · Program_Chairs · 2022-01-20

**Decision:**

Reject

**Comment:**

The paper explores the usefulness of intermediate layers for linear probing, aiming at improving out-of-distribution transfer with significantly less cost than fine-tuning. Two reviewers recommended borderline acceptance, while two others recommended borderline rejection as final rating. The main concerns raised by the reviewers were the limited novelty of the proposed method (e.g., compared to Elmo), unconvincing results in the natural and structure categories of VTAB, and lack of experiments to justify the claims, as well as the demonstration of the method in other tasks beyond image classification. The rebuttal has clarified several other questions. The AC really likes the simplicity of the approach, and also finds the problem of improving the efficiency of transfer learning very important. In addition, the paper is very well-written and easy to follow, as acknowledged by all reviewers. However, the AC agrees with R2 and R3 that the paper, in its current form, does not pass the bar of ICLR, unfortunately. First, the novelty is limited, as pointed out by R1, R2, and R3. In addition to the related works mentioned in the reviews like Elmo, note that the idea of selecting intermediate features, concatenating them, and running a linear classifier for OOD transfer has also been explored in [Yunhui Guo et al, A broader study of cross-domain few-shot learning, ECCV 2020]. Second, while the approach has advantages in terms of efficiency, the accuracy drop (compared to fine-tuning) for in-domain tasks limits its applicability. Finally, even though the AC agrees with the authors this is not a requirement, a more comprehensive set of experiments on more tasks would make the paper stronger, especially given that the novelty is incremental. The authors are encouraged to improve the paper for another top conference.